# Low-crystalline iron oxide hydroxide nanoparticle anode for high-performance supercapacitors

Kwadwo Asare Owusu[1,*], Longbing Qu[1,2,*], Jiantao Li[1], Zhaoyang Wang[1], Kangning Zhao[1], Chao Yang[1], Kalele Mulonda Hercule[3], Chao Lin[1], Changwei Shi[1], Qiulong Wei[1], Liang Zhou[1] & Liqiang Mai[1]

Carbon materials are generally preferred as anodes in supercapacitors; however, their low capacitance limits the attained energy density of supercapacitor devices with aqueous electrolytes. Here, we report a low-crystalline iron oxide hydroxide nanoparticle anode with comprehensive electrochemical performance at a wide potential window. The iron oxide hydroxide nanoparticles present capacitances of 1,066 and 716 F g$^{-1}$ at mass loadings of 1.6 and 9.1 mg cm$^{-2}$, respectively, a rate capability with 74.6% of capacitance retention at 30 A g$^{-1}$, and cycling stability retaining 91% of capacitance after 10,000 cycles. The performance is attributed to a dominant capacitive charge-storage mechanism. An aqueous hybrid supercapacitor based on the iron oxide hydroxide anode shows stability during float voltage test for 450 h and an energy density of 104 Wh kg$^{-1}$ at a power density of 1.27 kW kg$^{-1}$. A packaged device delivers gravimetric and volumetric energy densities of 33.14 Wh kg$^{-1}$ and 17.24 Wh l$^{-1}$, respectively.

[1] State Key Laboratory of Advanced Technology for Materials Synthesis and Processing, International School of Materials Science and Engineering, Wuhan University of Technology, Wuhan 430070, China. [2] Department of Mechanical and Aerospace Engineering, Monash University, Melbourne, VIC 3800, Australia. [3] Department of Chemistry, University of Kinshasa, No. 1 University Street, BP. Kinshasa IX, Democratic Republic of the Congo. * These authors contributed equally to this work. Correspondence and requests for materials should be addressed to L.Z. (email: liangzhou@whut.edu.cn) or to L.M. (email: mlq518@whut.edu.cn).

D ue to the fast depletion of fossil fuels and global warming, there is an urgent need for clean energy technologies to supplement and replace the conventional energy sources. At the forefront of clean energy technologies are high-performance energy storage devices, which are needed for the next-generation consumer electronics, biomedical devices and hybrid electric vehicles[1–3]. As a well-known energy storage device, the supercapacitor has attracted tremendous research attention recently due to its high power density ($1–10 \, kW \, kg^{-1}$), fast charge and discharge rate (within seconds) and long cycling life ($>100,000$ cycles)[4–7]. Electrical double-layer capacitor (EDLC) materials have been widely used in supercapacitors due to their large specific surface area, high electrical conductivity and low cost[8–11]. Although high power density and cycling stability have been realized by these materials, the attained capacitance and energy density are typically low[12–14]. This is because of the charge-storage mechanism for EDLC materials, which is dominated by charge separation at the electrode/electrolyte interface[14,15]. Pseudocapacitor materials can provide a higher capacitance than EDLC materials due to their surface/near-surface redox reactions[1,4,16–18].

Currently, the research on supercapacitor is focused on increasing the energy density while retaining comparable high power density[19]. Asymmetric and hybrid supercapacitors (HSCs) have been extensively studied as a promising strategy to increase the energy density[20–26]. A typical HSC consists of both faradaic and capacitive electrodes[12,27]. This design results in high energy density due to the contributions from the different charge-storage mechanisms and the extended operating potential window in aqueous electrolytes (up to 2 V)[28,29]. Faradaic cathode materials have been extensively studied leading to the development of high-performance cathodes for aqueous supercapacitors[20,21,30–32]. For instance, nickel-based oxides have been explored due to their improved electronic conductivity and rich redox reactions, arising from the high electrochemical activity of Ni[26,28,33,34]. Despite the high performance of these cathode materials, the maximum energy density of their hybrid cells in aqueous electrolytes is largely hindered by the low specific capacitance of commonly used carbon anodes[35–37]. Recently, crystalline iron oxides ($Fe_2O_3$, $Fe_3O_4$) and iron oxide hydroxide (FeOOH) have been studied as supercapacitor or battery-type anode materials due to their high theoretical capacitance, wide operating potential window, low cost and natural abundance[38–44]. Even though significant progress has been achieved for these materials, most of them exhibit short cycle life and poor rate performance. Low-crystalline or amorphous metal oxides are capable of achieving better cycling stability than the high-crystalline counterpart because of their more structural defects and disorder[30,45–47]. As far as we know, it is still a tremendous challenge to obtain anode materials with high capacitance, good rate capability and excellent cycling stability.

In the present work, we report a capacitive dominant FeOOH nanoparticle anode with comprehensive electrochemical performance at a wide potential window. The synthesis of the FeOOH nanoparticle anode involves the hydrothermal growth of iron oxide ($\alpha$-$Fe_2O_3$) nanoparticles on carbon fibre cloth (CFC) and the subsequent electrochemical transformation to low-crystalline FeOOH nanoparticles. The FeOOH anode manifests high specific capacitances at both low and high mass loadings, good rate capability (74.6% capacitance retention at $30 \, A \, g^{-1}$) and excellent cycling stability (91% capacitance retention after 10,000 cycles). To further evaluate the performance of the FeOOH nanoparticle anode for aqueous HSCs, we also designed the suitable battery-type cathode, nickel molybdate ($NiMoO_4$) using a hydrothermal method. An $NiMoO_4$//FeOOH aqueous hybrid device displays high specific capacitance ($273 \, F \, g^{-1}$), high energy density ($104.3 \, Wh \, kg^{-1}$) and exceptional stability. Importantly, a packaged device with an active material weight percentage of 35% shows high gravimetric and volumetric energy densities.

## Results

**Synthesis and characterization of $\alpha$-$Fe_2O_3$ nanoparticles**. We first synthesized $Fe_2O_3$ nanoparticles on CFC substrate through a facile hydrothermal method (Supplementary Fig. 1). Fig. 1a shows the X-ray diffraction (XRD) pattern of the $Fe_2O_3$. The XRD pattern can be indexed to rhombohedral $\alpha$-$Fe_2O_3$ (JCPDS card no. 00-033-0664) with R-3c space group and lattice parameters of $a = b = 5.0356 \, Å$ and $c = 13.7500 \, Å$. The $\alpha$-$Fe_2O_3$ sample was further characterized by Raman spectroscopy (Fig. 1b). A distant band is located at $1,316 \, cm^{-1}$ and the narrow bands located at 221 and $492 \, cm^{-1}$ can be assigned to the $A_{1g}$ modes, while the bands located at 247, 291, 407 and $607 \, cm^{-1}$ are due to the $E_{1g}$ modes of $\alpha$-$Fe_2O_3$[48,49]. The Raman spectrum confirms the existence of $\alpha$-$Fe_2O_3$. The surface area of the $\alpha$-$Fe_2O_3$ was also studied by nitrogen sorption (Supplementary Fig. 2a). The Brunauer–Emmett–Teller (BET) surface area of the $\alpha$-$Fe_2O_3$ is determined to be $41 \, m^2 \, g^{-1}$.

The morphology of $\alpha$-$Fe_2O_3$ was identified with scanning electron microscopy (SEM) and transmission electron microscopy (TEM). As shown in Fig. 1c, uniformly distributed nanoparticle morphology can be observed. The SEM image at a higher magnification (Fig. 1c, inset) reveals that the nanoparticles are uniform in size and strongly attached to the CFC substrate. From the TEM image (Fig. 1d), the diameter of the nanoparticles is determined to be ~30 nm. The high-resolution TEM (HRTEM) image of the $\alpha$-$Fe_2O_3$ nanoparticles is shown in Fig. 1e. Lattice fringes with interplanar spacing of 0.36 nm corresponding to the (0 1 2) plane of $\alpha$-$Fe_2O_3$ can be clearly discerned. The polycrystalline feature of the $\alpha$-$Fe_2O_3$ nanoparticles is confirmed by the selected area electron diffraction (SAED) pattern (Fig. 1f). It shows a set of concentric rings, which can be indexed to the (104), (113), (116) and (300) diffractions of rhombohedral $\alpha$-$Fe_2O_3$.

**Transformation into low-crystalline FeOOH nanoparticles**. The $\alpha$-$Fe_2O_3$ is transformed into low-crystalline FeOOH during electrochemical cycles in the potential range between $-1.2$ and 0 V versus saturated calomel electrode (SCE) (Supplementary Fig. 1). The cyclic voltammetry (CV) curves of the $\alpha$-$Fe_2O_3$ electrode at different cycles in 2 M KOH are shown in Fig. 2a. A pair of faradaic peaks positioned at $-0.66$ and $-1.05$ V versus SCE is observed during the first cycle. The intensity of the peaks gradually reduces during the first ten cycles (defined as activation process) and becomes stable afterwards, which suggests that some changes in crystalline structure have occurred during the first ten cycles. The CV curves after the activation process portray a quasirectangular shape with very broad peaks. To understand the structure changes and charge-storage mechanism of the anode, *ex situ* XRD, X-ray photoelectron spectroscopy (XPS), SEM and TEM tests were carried out. As shown in Fig. 2b, the $\alpha$-$Fe_2O_3$ is transformed into FeOOH (JCPDS No. 01-077-0247) after ten electrochemical cycles. The $\alpha$-$Fe_2O_3$ phase cannot be recovered in the subsequent discharge process, instead, a mixture of FeOOH and $Fe(OH)_2$ is obtained. SEM images of the transformed FeOOH show that the nanoparticle morphology is well maintained (Supplementary Fig. 3). Also, the TEM and HRTEM images (Fig. 2c,d) further confirm that the $\alpha$-$Fe_2O_3$ is transformed into low-crystalline FeOOH nanoparticles during the activation process. XPS test was carried out to confirm the valence states of the various elements on the surface of $\alpha$-$Fe_2O_3$ after activation. The Fe $2p$ core-level spectrum (Fig. 2e) shows two characteristic

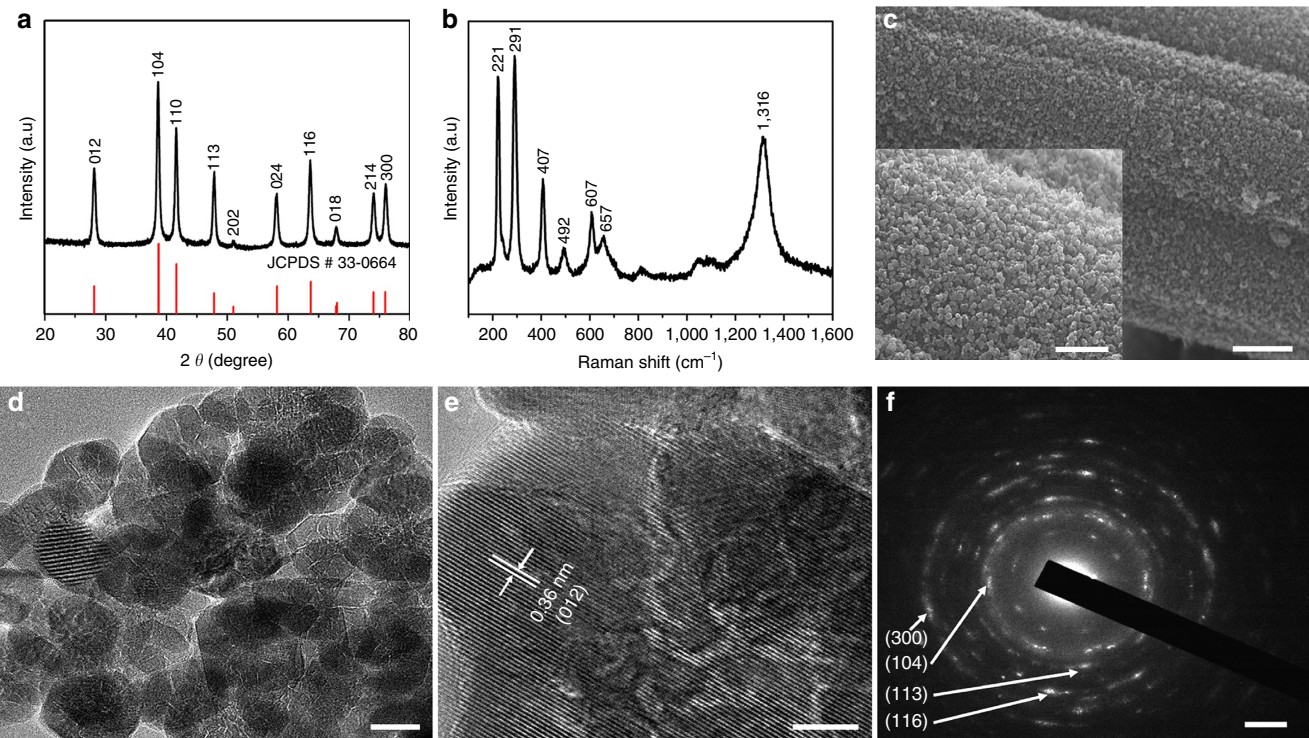

**Figure 1 | Material characterization of α-Fe₂O₃ nanoparticles.** (**a**) XRD pattern. (**b**) Raman spectrum. (**c**) SEM images. Scale bars, 1 μm (inset: 500 nm). (**d**) TEM image. Scale bar, 20 nm. (**e**) HRTEM image. Scale bar, 5 nm. (**f**) SAED pattern. Scale bar, 2 nm⁻¹.

peaks located at 711 and 725 eV corresponding to Fe $2p_{1/2}$ and Fe $2p_{3/2}$ spin orbitals of FeOOH, together with two satellite peaks at 717 and 733 eV[46,47]. The deconvolution of the O $1s$ core-level spectrum (Fig. 2f) shows three distinct oxygen contributions corresponding to H-O-H (532.7 eV), Fe-O-H (531.4 eV) and Fe-O-Fe bonds (530.4 eV)[46,47,50]. The H-O-H bond corresponds to water molecule, which suggests that the FeOOH nanoparticles are in hydrated form[50]. The XPS characterization confirms the electrochemical transformation of α-Fe₂O₃ nanoparticles to FeOOH nanoparticles, which is highly consistent with the *ex situ* XRD, SEM and TEM results. According to the above characterizations, the probable transformation reaction and charge-storage mechanism is proposed as follows:

$$\text{The activation process}: \text{Fe}_2\text{O}_3 + \text{H}_2\text{O} \rightarrow 2\text{FeOOH} \quad (1)$$

$$\begin{aligned} \text{Subsequent discharge process}: \text{FeOOH} + \text{H}_2\text{O} + \text{e}^- \\ \rightarrow \text{Fe(OH)}_2 + \text{OH}^- \end{aligned} \quad (2)$$

$$\begin{aligned} \text{Subsequent charge process}: \text{Fe(OH)}_2 + \text{OH}^- \\ \rightarrow \text{FeOOH} + \text{H}_2\text{O} + \text{e}^- \end{aligned} \quad (3)$$

**Electrochemical performance of FeOOH nanoparticles.** To study the electrochemical performance of the low-crystalline FeOOH nanoparticles, CV and galvanostatic charge/discharge tests were carried out in a three-electrode system with a Pt plate counter-electrode and an SCE reference electrode in 2 M KOH electrolyte. Fig. 3a displays the CV curves of the FeOOH nanoparticles tested at different scan rates ranging from 5 to 50 mV s⁻¹ in a − 1.2 to 0 V versus SCE potential window. The quasirectangular shape CV curves of the FeOOH nanoparticle anode denote an electrochemical signature of a typical pseudocapacitive electrode[12,27]. The symmetric CV curves also indicate that the charge storage process and the redox reactions are

reversible. The charge/discharge curves of the FeOOH nanoparticles are shown in Supplementary Fig. 4a. The specific gravimetric and areal capacitances of the FeOOH nanoparticles are calculated from the discharge curves. As displayed in Fig. 3b and Supplementary Fig. 4b, the FeOOH nanoparticles exhibit a capacitance of 1,066 F g⁻¹ (1.71 F cm⁻²) at 1 A g⁻¹. With the increase of the current density to 30 A g⁻¹, a capacitance of 796 F g⁻¹ (1.27 F cm⁻²) can be maintained, corresponding to 74.6% of the capacitance at 1 A g⁻¹. Another important performance metric in characterizing supercapacitor electrodes is the mass loading of the active materials[51]. Considering the mass loading of typical industrial porous carbon electrodes (∼ 10 mg cm⁻²), we tuned the mass loading of the FeOOH anode. The FeOOH anode displays quasirectangular-shaped CV curves and symmetric triangular charge/discharge curves irrespective of the mass loading (Supplementary Fig. 5). With mass loadings of 1.6, 3.0, 5.6 and 9.1 mg cm⁻², the low-crystalline FeOOH nanoparticle anode displays specific gravimetric capacitances of 1,066, 996, 827 and 716 F g⁻¹ at 1 A g⁻¹, respectively (Fig. 3c). The capacitances of the FeOOH anode decrease with increasing mass loadings; however, they still exhibit good rate capabilities (Supplementary Fig. 6b). The areal and volumetric capacitances of the FeOOH anode (including the volume of the current collector) with a high mass loading of 9.1 mg cm⁻² can reach as high as 6.5 F cm⁻² (Fig. 3c) and 186 F cm⁻³ (Fig. 3d).

As one of the main parameters for supercapacitors, the long-term cycling stability of the FeOOH anode was studied (Fig. 3e). For the FeOOH anode with a mass loading of 1.6 mg cm⁻², 91% of the initial capacitance can be retained after 10,000 charge/discharge cycles at 30 A g⁻¹, whereas 86% of the initial capacitance is retained for the anode with a mass loading of 9.1 mg cm⁻² after 10,000 cycles at 15 A g⁻¹. At 1 A g⁻¹, the FeOOH electrode displays a low voltage drop of 0.0097 V, suggesting a low internal resistance ($R_s$) of the electrode

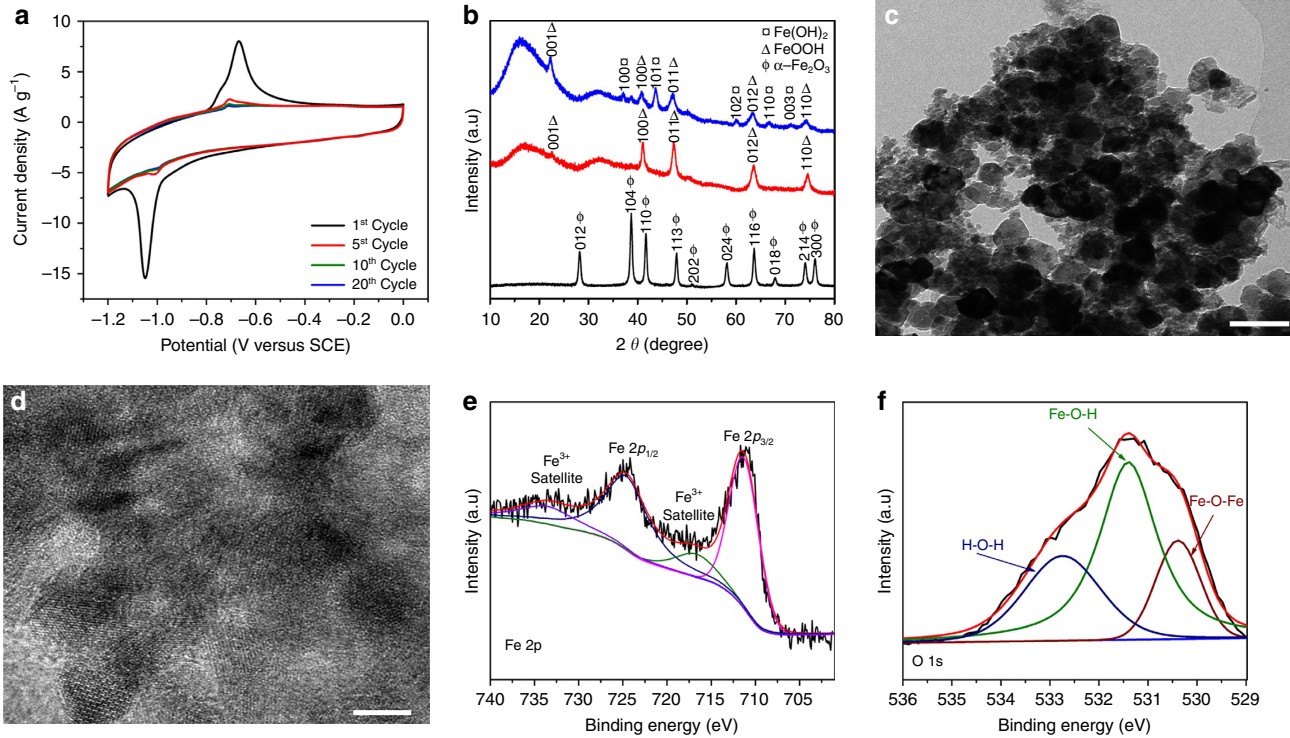

**Figure 2 | Transformation of α-Fe₂O₃ into FeOOH nanoparticles. (a)** CV curves of α-Fe₂O₃ electrode at $2\,mV\,s^{-1}$. **(b)** XRD patterns of the electrodes before activation (black line), after activation in charge state (red line), and after activation in discharge state (blue line). **(c)** TEM image of FeOOH nanoparticles. Scale bar, 100 nm. **(d)** HRTEM images of FeOOH nanoparticles. Scale bar, 5 nm. **(e,f)** Fe2p and O 1s XPS core-level spectra of FeOOH nanoparticles.

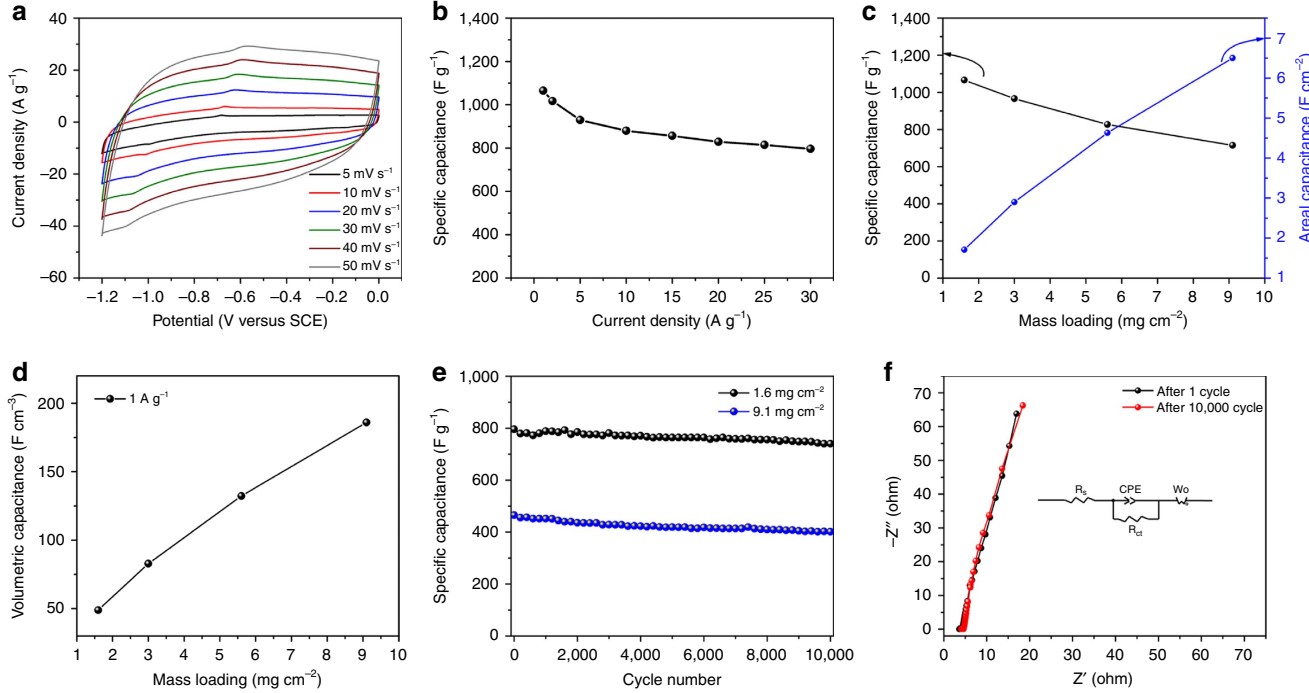

**Figure 3 | Electrochemical performance of FeOOH nanoparticle anode. (a)** CV curves. **(b)** Specific gravimetric capacitance as a function of current density. **(c)** Specific gravimetric and areal capacitances of the FeOOH nanoparticle anode at different mass loadings. **(d)** Volumetric capacitance of the FeOOH nanoparticle anode (including the volume of the current collectors) at different mass loadings. **(e)** Cycling performance of the FeOOH nanoparticle anode at 1.6 and $9.1\,mg\,cm^{-2}$. **(f)** Nyquist plot after 1st and 10,000th cycle.

$(3.45\,\Omega)^{52}$. The $R_s$ and charge transfer resistance ($R_{ct}$) after the first cycle obtained from the simulation of the Nyquist plot are 3.59 and $0.59\,\Omega$, respectively (Fig. 3f). After 10,000 charge/ discharge cycles, the $R_s$ increases to $4.10\,\Omega$, whereas the $R_{ct}$ reduces to $0.50\,\Omega$ (Supplementary Table 1). The reduced $R_{ct}$ suggests that the low-crystalline FeOOH facilitates fast

diffusion of electrolyte ions, advantageous to the long-term stability of the anode[53].

**Synthesis and characterization of NiMoO$_4$ nanowires.** The synthesis of NiMoO$_4$ nanowire cathode was achieved through a hydrothermal method followed by postannealing. The crystallographic phase of NiMoO$_4$ was characterized by XRD analysis. Fig. 4a depicts the Rietveld refined XRD pattern of the NiMoO$_4$. The lattice parameters of NiMoO$_4$ ($a = 9.5982$ Å, $b = 8.7760$ Å and $c = 7.6717$ Å) calculated by Rietveld refinement match well with monoclinic NiMoO$_4$ (JCPDS No. 01-086-0361; $a = 9.5660$ Å, $b = 8.7340$ Å and $c = 7.6490$ Å). The weighted profile Rietveld factor ($R_{wp}$) of the NiMoO$_4$ is determined to be 6.853% (Supplementary Table 2). XPS was applied to verify the surface composition of the NiMoO$_4$ nanowires (Supplementary Fig. 7a). The Ni $2p$ core-level spectrum shows two major peaks with binding energies of 856.19 and 873.92 eV, corresponding to Ni $2p_{1/2}$ and Ni $2p_{3/2}$ of Ni$^{2+}$, respectively (Supplementary Fig. 7b)[54,55]. The Mo $3d$ core-level spectrum presents two characteristic peaks with binding energies of 232.36 and 235.5 eV, corresponding to Mo $3d_{5/2}$ and Mo $3d_{3/2}$ of Mo$^{6+}$, respectively (Supplementary Fig. 7c)[56,57]. Last, the deconvolution of O $1s$ core-level spectrum shows two major oxygen contributions (Supplementary Fig. 7d). The peak located at 530.4 eV is associated with the metal-oxygen bond, while the peak at 531.4 eV corresponds to the lattice oxygen[57].

The morphology of NiMoO$_4$ grown on nickel foam substrate was observed with SEM and TEM. From the low-magnification SEM (Fig. 4b and inset), it can be easily observed that the nanowires are grown on the surface of the nickel foam. A high-magnification SEM (Fig. 4c) shows that the bundled nanowires have needle-like tips. The presence of spaces between adjacent nanowires would enhance the penetration of the electrolyte ions[28,58]. TEM image of a typical NiMoO$_4$ nanowire is shown in Fig. 4d. The diameter of the NiMoO$_4$ nanowires is determined to be 50–100 nm. The HRTEM image of the NiMoO$_4$ nanowire is shown in Fig. 4e, from where the (0 2 0) lattice fringes with a lattice spacing of 0.43 nm is clearly observed. The polycrystallinity of the NiMoO$_4$ nanowires is confirmed from the SAED pattern (Fig. 4f), as it shows Bragg spots corresponding well with (-205), (2 0 4), (-113), (111) and (-313) planes of monoclinic NiMoO$_4$. The NiMoO$_4$ displays a type II isotherm with an H3 hysteresis loop (Supplementary Fig. 2b), and the BET surface area is determined to be 49 m$^2$ g$^{-1}$.

**Electrochemical performance of NiMoO$_4$ nanowires.** Fig. 5a shows the CV curves of NiMoO$_4$ at different scan rates from 1 to 10 mV s$^{-1}$ tested between 0 and 0.5 V versus SCE. From the linear sweep voltammetry (LSV) analysis (Supplementary Fig. 8f), it can be observed that oxygen evolution starts at ~0.52 V versus SCE in the NiMoO$_4$ electrode. Thus, it is safe for NiMoO$_4$ to be cycled between 0 and 0.5 V versus SCE. The charge-storage mechanism in NiMoO$_4$ can be ascribed to faradaic battery-type mechanism from the sharp peaks of the CV curves[12,27]. The curves show a pair of anodic and cathodic peaks arising from the fast faradaic redox reactions of Ni(II) ↔ Ni(III) during charge and discharge[28,55,56]. The NiMoO$_4$ nanowires exhibit very good electrochemical reversibility as evidenced by the near mirror symmetry of both anodic and cathodic peaks[55]. The specific capacity instead of specific capacitance of the NiMoO$_4$ cathode was calculated from the discharge curves (Supplementary Fig. 9a) to give realistic values of the energy storage and release[12,27,36]. As shown in Fig. 5b and Supplementary Fig. 9b, the NiMoO$_4$ electrode delivers a specific capacity of 223 mAh g$^{-1}$ (0.33 mAh cm$^{-2}$) at 1 A g$^{-1}$ and 59% of the capacity can be retained at 30 A g$^{-1}$ (130 mAh g$^{-1}$, 0.2 mAh cm$^{-2}$). The long-term cycling performance of the NiMoO$_4$ nanowires was also studied. The NiMoO$_4$ nanowires display capacitance retention of 85.1% after 10,000 charge/discharge cycles at 30 A g$^{-1}$ (Fig. 5c).

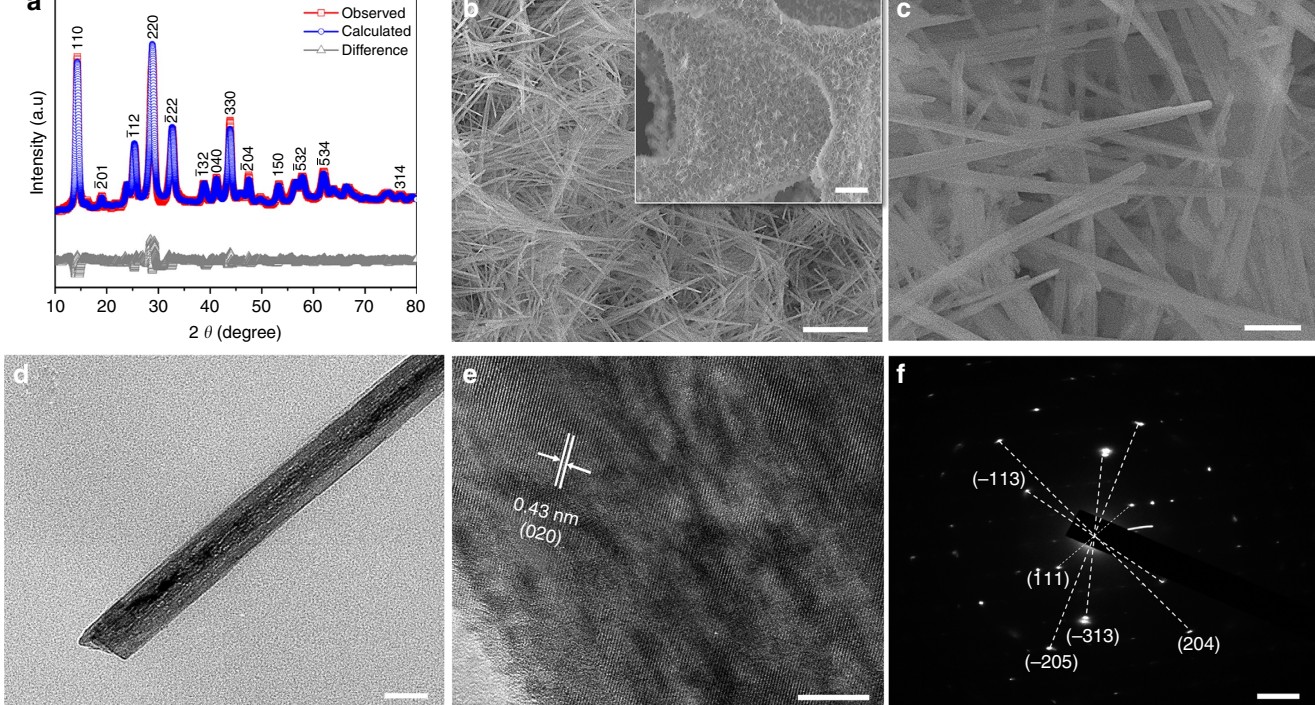

**Figure 4 | Material characterization of NiMoO$_4$ nanowires.** (**a**) Rietveld refined XRD pattern. (**b**) Low magnification SEM images. Scale bar, 3 μm (inset 30 μm). (**c**) High magnification SEM image. Scale bar, 300 nm. (**d**) TEM image. Scale bar, 50 nm. (**e**) HRTEM image. Scale bar, 10 nm. (**f**) SAED pattern. Scale bar, 2 nm$^{-1}$.

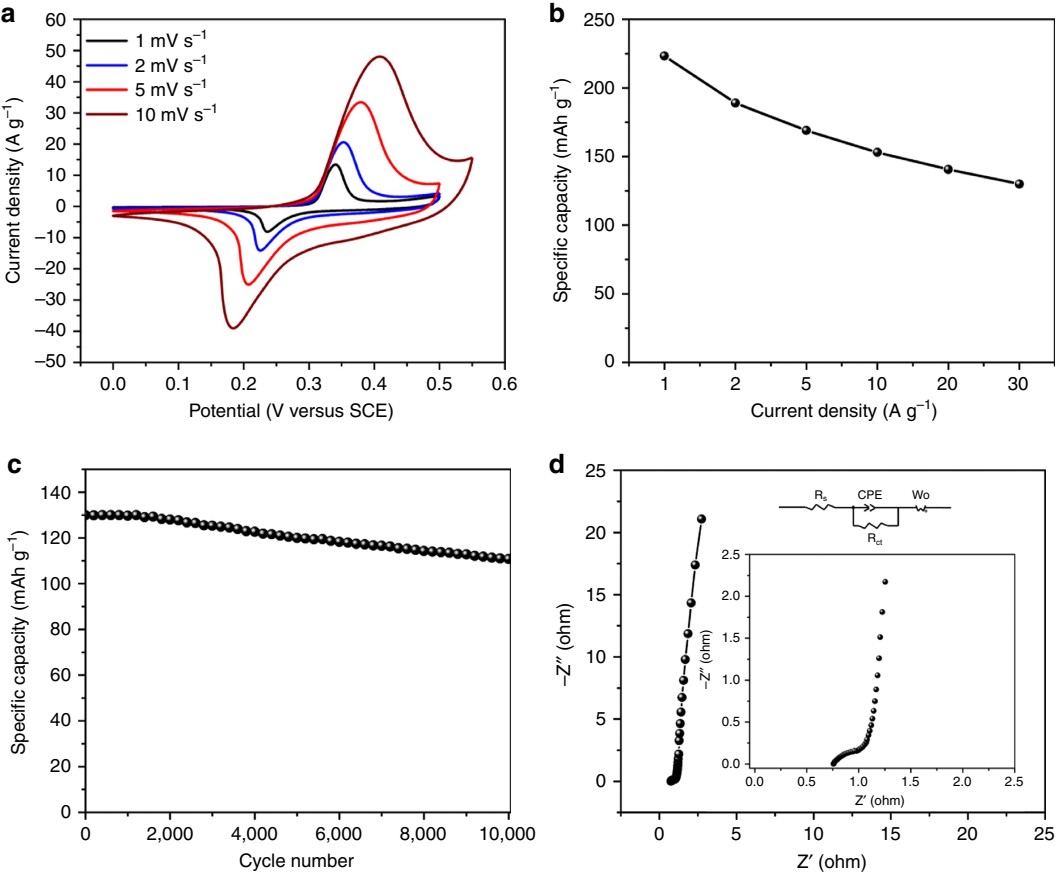

**Figure 5 | Electrochemical performance of the NiMoO$_4$ nanowire cathode.** (**a**) CV curves. (**b**) Specific capacity as a function of current density. (**c**) Cycling performance at 30 A g$^{-1}$. (**d**) Nyquist plot, inset is the magnified view of the Nyquist plot in high-frequency region.

To provide further insights, EIS was measured to quantify the resistance at the electrode/electrolyte interface (Fig. 5d). The NiMoO$_4$ nanowires display $R_s$ and $R_{ct}$ values of 0.72 and 0.15 Ω, respectively.

**Electrochemical evaluation of aqueous HSC.** To further evaluate the practical application of the FeOOH anode, an aqueous HSC was assembled with the NiMoO$_4$ and FeOOH as the cathode and anode, respectively. The NiMoO$_4$ cathode and FeOOH anode are mass balanced at 5.5 A g$^{-1}$ (Supplementary Fig. 10). As shown in Fig. 6a, series of CV tests are undertaken in different potential windows in 2 M KOH to determine the optimal operating potential window of the NiMoO$_4$/FeOOH HSC. Under a potential window of 1.1 V, only one anodic peak is visible, implying that there is no contribution from the cathode and the reactions are irreversible (Fig. 6a). Under a wide potential window of 1.9 V, the aqueous electrolyte begins to decompose. The optimal potential window of the assembled HSC is determined to be 1.7 V. This is in good agreement with the working potential windows of the separate electrodes with respect to the water oxidation and reduction potentials in 2 M KOH electrolyte (Supplementary Fig. 8). With the increase of voltage potential from 1.1 to 1.7 V at 11.25 A g$^{-1}$, the capacitance increases from 87.05 to 230.72 F g$^{-1}$ (Supplementary Fig. 11a), which is mainly due to the increased redox reactions of the electrodes and it can be confirmed from the CV integral area (Fig. 6a). Fig. 6b displays typical CV curves of the HSC at different scan rates in a 1.7 V potential window. The CV curves have a non-rectangular shape with a couple of broad reversible

redox peaks, which indicate the capacitance mainly comes from the redox reactions. The galvanostatic charge/discharge curves of the NiMoO$_4$//FeOOH HSC at different current densities were tested (Supplementary Fig. 11b). As shown in Fig. 6c, the full HSC delivers a specific capacitance of 273 and 183 F g$^{-1}$ at a current density of 1.5 and 22.5 A g$^{-1}$, respectively. The HSC device displays good rate capability with 67% of the capacitance retained in that current density range. Supplementary Fig. 11c shows the long-term cycling stability of the NiMoO$_4$//FeOOH HSC and it retains 80.8% of its initial specific capacitance after 10,000 cycles at a current density of 22.5 A g$^{-1}$. The float voltage test, a more demanding test than the conventional charge/discharge cycling was also used to study the stability of the NiMoO$_4$//FeOOH HSC in 2 M KOH electrolyte[59,60]. For a test time of 450 h, the NiMoO$_4$//FeOOH HSC displays exceptional stability with no loss in capacitance (Fig. 6d).

The energy and power density of the HSC were calculated from the galvanostatic discharge curves and plotted in the Ragone plot (Fig. 6e). The HSC displays a maximum gravimetric energy density of 104.3 Wh kg$^{-1}$ at a power density of 1.27 kW kg$^{-1}$ and an energy density of 31 Wh kg$^{-1}$ at a maximum power density of 10.94 kW kg$^{-1}$. Volumetric capacitance, volumetric energy and power density are very important parameters for practical applications of supercapacitors[51]. The NiMoO$_4$//FeOOH packaged device displays high volumetric capacitances; even though the active material mass accounts for just 6.5 wt% of the packaged device, the volumetric capacitances still reach 8.24 and 5.53 F cm$^{-3}$ at 1.5 and 22.5 A g$^{-1}$, respectively (Fig. 6c). The HSC device also displays a maximum volumetric energy density of 3.15 mWh cm$^{-3}$ at a power density

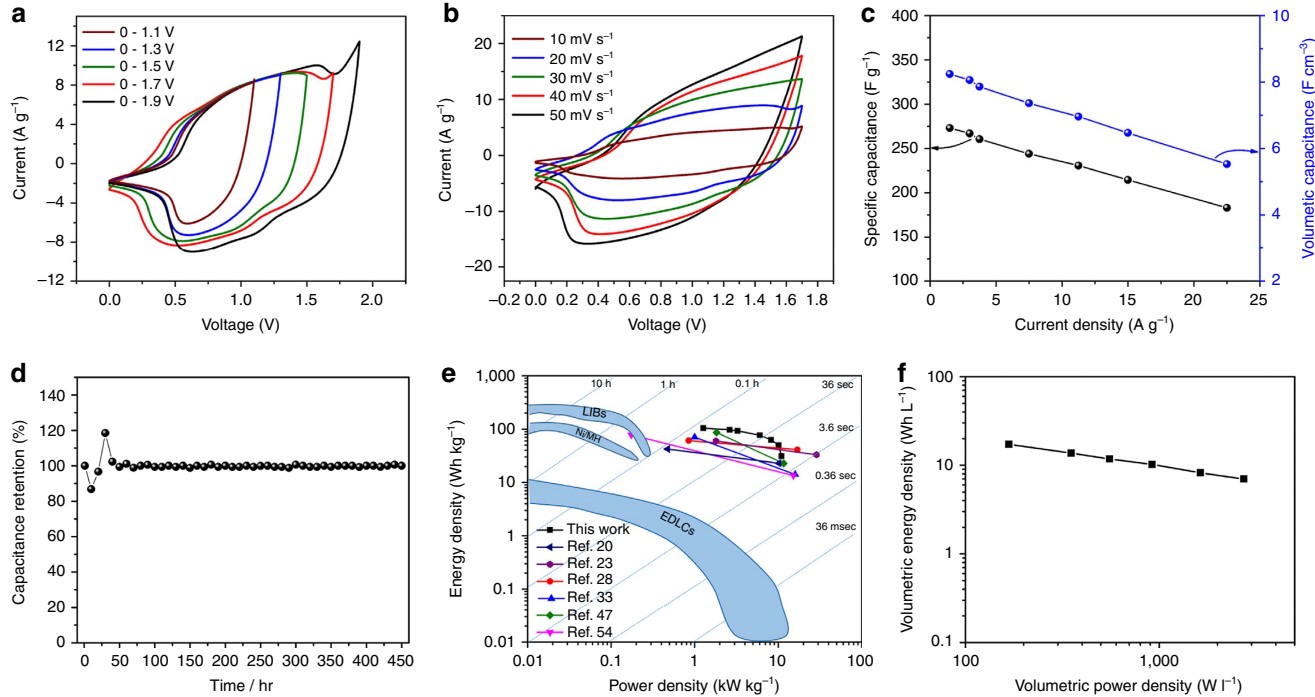

**Figure 6 | Electrochemical performance of NiMoO₄//FeOOH HSC.** (**a**) CV curves in different potential windows at 20 mV s⁻¹. (**b**) CV curves of the HSC device at various scan rates from 10 to 50 mV s⁻¹ in 1.7 V potential window. (**c**) The specific gravimetric and volumetric capacitances of the HSC at different current densities. (**d**) Float voltage stability test of the NiMoO₄//FeOOH HSC for 450 h. (**e**) Ragone plot of the NiMoO₄//FeOOH HSC, the energy and power densities of recently reported Ni-metal oxide-based SCs and the conventional storage devices are added for comparison. (**f**) Volumetric energy and power density of the NiMoO₄//FeOOH packaged device. Active material mass accounts for 35% of the total packaged weight.

of 38.33 mW cm⁻³ and a maximum volumetric power density of 330.62 mw cm⁻³ at an energy density of 0.68 mWh cm⁻³ (Supplementary Fig. 12). For practical applications, a NiMoO₄//FeOOH packaged device with active materials accounting for 35% of the total weight is also assembled. It displays a volumetric capacitance of 42.96 F cm⁻³, a maximum energy density of 31.44 Wh kg⁻¹ at a power density of 305 W kg⁻¹ and a maximum power density of 4,976 W kg⁻¹ at an energy density of 12.72 W kg⁻¹ (Supplementary Fig. 13). Last, the packaged device displays maximum volumetric energy and power densities of 17.24 Wh l⁻¹ and 2,736.08 W l⁻¹, respectively (Fig. 6f).

## Discussion

Using *ex situ* XRD, XPS, SEM and TEM tests, it has been unambiguously demonstrated that not only the surface but also the bulk of the $\alpha$-$Fe_2O_3$ nanoparticles can be converted into low-crystalline FeOOH during the electrochemical activation process, which has been rarely reported. The FeOOH anode shows characteristic capacitive CV profiles with broad peaks indicating that the stored charge is mainly pseudocapacitive[61]. From the CV curves, the capacitive ($k_1$) and diffusion ($k_2$)-controlled contributions to the total capacity at a particular voltage can be separated using the equation shown below[61-63]:

$$i(V) = k_1 v + k_2 v^{\frac{1}{2}} \tag{4}$$

where $v$ is the sweep rate. Fig. 7a–c show a typical separation of capacitive and diffusion currents at scan rates of 1, 2 and 5 mV s⁻¹, respectively. As shown in Fig. 7d, the capacitive-controlled process contributes 78.9%, 84.6% and 89.6% of the total charge storage at 1, 2 and 5 mV s⁻¹, respectively, suggesting the dominant capacitive charge-storage mechanism in the FeOOH anode. The dominant capacitive storage endows extraordinary high charge storage kinetics and stable cycling

performance[53,61-63]. As a result, even at high mass loadings of ∼5.6 and 9.1 mg cm⁻², the FeOOH anode exhibits excellent comprehensive electrochemical performances, which are essential for the practical application of supercapacitors. Compared to carbon materials, the high specific capacitance of the low-crystalline FeOOH nanoparticles validates its selection as the anode for fabricating the full HSC[35,36]. To the best of our knowledge, the low-crystalline FeOOH nanoparticles display superior electrochemical performances to previously reported iron oxide based nanostructured electrodes (Supplementary Table 3). The FeOOH anode presents the advantages of a wide operating potential window, dominant capacitive charge-storage mechanism and the low-crystalline feature in comparison to several reported iron oxides, which are diffusion-controlled (Supplementary Table 3). A plot of the voltage drops versus current density of both electrodes displays very gentle slopes, which can be ascribed to the low internal resistances and excellent conductivities of the electrodes (Supplementary Fig. 14). The very steep slopes in the Warburg region (Figs 3f and 5d) indicate high ion mobility and diffusion, which is favourable for rate capability and cycling stability.

Compared with recently reported metal oxide//carbon material full supercapacitors, the NiMoO₄//FeOOH HSC displays superior specific capacitance[20-23,26,28]. Furthermore, the energy density of the assembled HSC exceeds recently reported nickel-based full supercapacitors, such as $Ni_2Co_2S_4$//G/CS paper (42.3 Wh kg⁻¹ at 476 W kg⁻¹)[20], Ni(OH)₂/graphene//porous graphene (77.8 Wh kg⁻¹ at 174.7 W kg⁻¹)[54], FeOOH//Co-Ni-DH (86.4 Wh kg⁻¹ at 1.83 kW kg⁻¹)[47], NiMoO₄//activated carbon (60.9 Wh kg⁻¹ at 850 W kg⁻¹)[28], Ni-Co-S//graphene film (60 Wh kg⁻¹ at 1.8 kW kg⁻¹)[23] and NiMoO₄//NiMoO₄ (70.7 Wh kg⁻¹ at 1 kW kg⁻¹)[33]. The excellent electrochemical performance of the NiMoO₄//FeOOH HSC may be attributed to the following factors: (1) The dominant

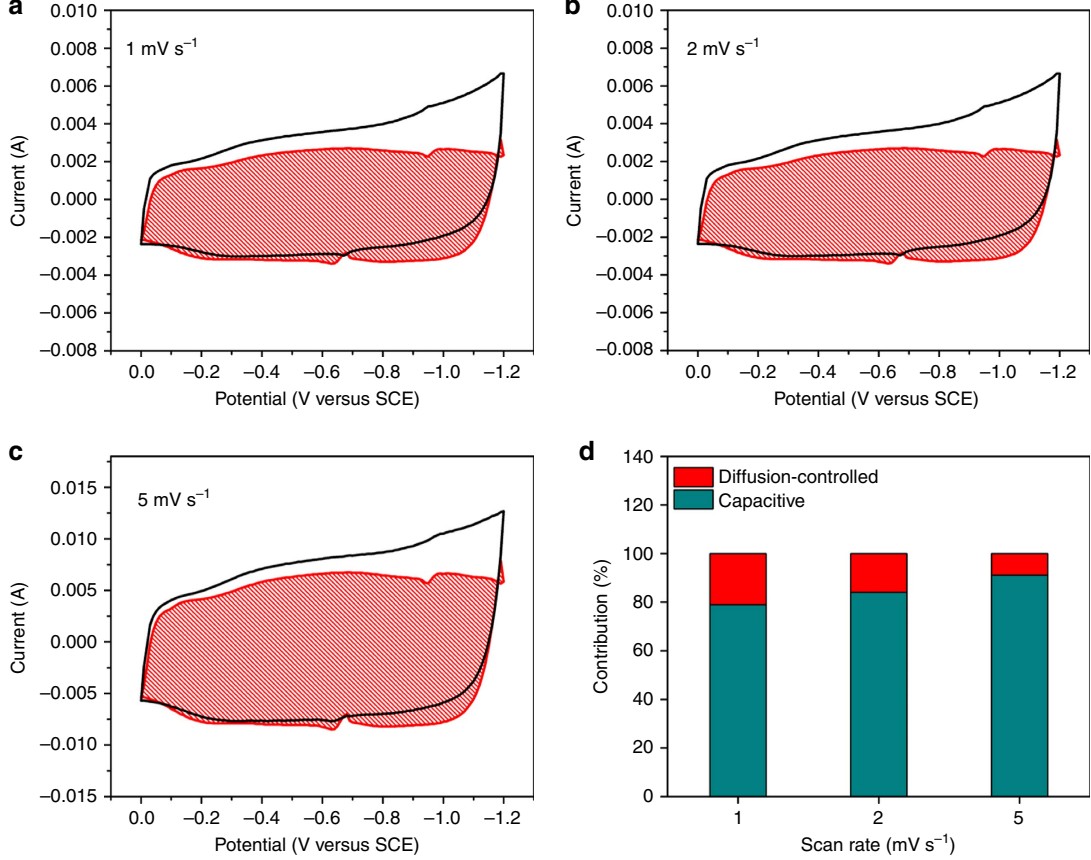

**Figure 7 | Capacitive and diffusion-controlled contributions to charge storage.** Voltammetric responses for low-crystalline FeOOH nanoparticles at scan rates of (**a**) 1, (**b**) 2 and (**c**) 5 mV s$^{-1}$. The capacitive contribution to the total current is shown by the shaded region. (**d**) The capacitance contribution at different scan rates (1, 2 and 5 mV s$^{-1}$).

capacitive contribution of the low-crystalline FeOOH nanoparticle anode results in high capacitances in a wide potential window, which translates into the high-energy density of the hybrid device. (2) The FeOOH nanoparticles present short ion diffusion paths, which are favourable for fast redox reactions, and the low crystalline structure has the self-adaptive strain-relaxation capability during the charge and discharge processes, leading to high stability. (3) The large surface area of the active materials provides more active sites for charge storage. (4) The direct growth of active materials on conductive substrates eliminates the use of a binder, which often inhibits electrode/electrolyte contact areas and increases the overall resistances of the electrodes. (5) The highly conductive and porous substrates (nickel foam and CFC) provide continuous electronic transport and easy accessibility of the electrolytes to the active materials.

In summary, we have successfully developed low-crystalline FeOOH nanoparticles through a novel strategy involving the hydrothermal growth and the subsequent electrochemical transformation of α-Fe$_2$O$_3$ nanoparticles. The low-crystalline design of the pseudocapacitive anode with high comprehensive performance largely enhances the energy and power densities of the supercapacitor. Therefore, the well-designed low-crystalline FeOOH materials could be a very suitable supercapacitor anode for future practical applications due to its low cost, easy preparation, environmental benignity and high comprehensive electrochemical performance at a wide potential window. An assembled NiMoO$_4$//FeOOH HSC displays a high capacitance of 273 F g$^{-1}$ at 1.5 A g$^{-1}$ and a high energy density of 104.3 Wh kg$^{-1}$ at a power density of 1.27 kW kg$^{-1}$ in an extended potential window (1.7 V), which largely overcomes the present tremendous challenge of the low-energy density of supercapacitors. Our work also provides a promising design direction for optimizing the electrochemical performance of full supercapacitors using various pseudocapacitive materials with suitable reaction potentials.

## Methods

**Synthesis of FeOOH nanoparticles.** Fe(NO$_3$)$_3$·9H$_2$O (1.212 g) was dissolved in 60 ml distilled water and stirred for 2 h. Afterwards, the resultant clear solution was transferred into a Teflon-lined stainless-steel autoclave containing precleaned CFC. The hydrothermal process was carried out at 120 °C for 24 h. After cooling, the substrate was removed and washed with distilled water. The sample was dried at 70 °C for 12 h to obtain the α-Fe$_2$O$_3$ nanoparticles. The mass of the α-Fe$_2$O$_3$ nanoparticles on the CFCs can be easily tuned by controlling the synthesis conditions. The α-Fe$_2$O$_3$ nanoparticles electrodes were electrochemically cycled in a three-electrode cell system by using the as-synthesized materials on the CFC substrate as the working electrode, Pt plate as counter-electrode and SCE as reference electrode in 2 M KOH electrolyte. The α-Fe$_2$O$_3$ nanoparticles electrodes were fully transformed into FeOOH nanoparticles after the tenth cycle in a − 1.2 to 0 V versus SCE potential window.

**Synthesis of NiMoO$_4$ nanowires.** The synthesis of NiMoO$_4$ nanowires was achieved by a mild hydrothermal method with postannealing. In a typical synthesis, NiCl$_2$·6H$_2$O (0.725 g) was dissolved in 25 ml H$_2$O and stirred for 1 h, followed by drop-wise addition of 25 ml aqueous solution of Na$_2$MoO$_4$·2H$_2$O (0.740 g) and stirred for another 2 h (all chemicals were used as received without purification). The resultant solution was transferred into a Teflon-lined stainless-steel autoclave containing precleaned nickel foams and kept at 120 °C for 12 h. The as-synthesized precursor was then ultrasonically cleaned at 50 Hz for 5 min in distilled water, dried at 70 °C overnight and finally annealed in argon at 400 °C for 2 h at a ramping rate of 5 °C min$^{-1}$.

**Characterization.** The crystallographic characterization of the as-synthesized samples was performed with a Bruker D8 Advance X-ray diffractometer with a

non-monochromatic Cu Ka X-ray source. Field emission SEM images were obtained with a JEOL-7100F microscope. TEM images were collected with a JEM-2100F STEM/EDS microscope. The BET surface area was measured using a Tristar II 3,020 instrument at 77 K. Raman spectrum was achieved using a Renishaw RM-1000 laser Raman microscopy system. XPS measurements were performed using a VG Multi Lab 2,000 instrument.

**Determination of mass loading.** The conductive substrates, nickel foam ($2 \times 5 \times 0.04$ cm$^3$) and CFCs ($2 \times 7 \times 0.035$ cm$^3$) were initially weighed before the growth of the active materials. All the samples were washed with distilled water and dried thoroughly at 80 °C overnight before being weighed with an analytical balance. The mass of the active materials was determined by the mass difference (before and after drying for the anode; before and after calcination for the cathode) divided by the macroscopic area of the conductive substrates. The mass loading of the NiMoO$_4$ and FeOOH is 1.5 and 1.6 mg cm$^{-2}$, respectively. The thickness of the samples was measured with a vernier caliper.

**Electrochemical measurements.** The electrochemical measurements of the individual electrode samples were carried out in a three-electrode cell system with the as-synthesized materials on the conductive substrates as the working electrode, SCE as reference electrode and Pt plate as counter-electrode in a 2 M KOH electrolyte using an electrochemical workstation (CHI 760D).

The specific capacitances of the electrodes and devices were calculated from the galvanostatic discharge curves at different current densities using the formula below.

$$C_s = \frac{I \times \Delta t}{m \times \Delta V} \tag{5}$$

where $C_s$ (F g$^{-1}$) is the specific capacitance, $I$ (A) is the discharge current, $\Delta t$ (s) is the discharge time, $m$ (g) is the mass of the active material and $\Delta V$ is the operating voltage (obtained from the discharge curves excluding the potential drop). The areal capacitance ($C_A$, F cm$^{-2}$) of the electrodes was calculated by replacing the mass loading with the surface area of the electrodes (1 cm$^2$). The volumetric capacitance ($C_v$, F cm$^{-3}$) of the FeOOH nanoparticle anode was calculated by replacing the mass of the active material with the volume of the electrodes (including the volume of the current collectors).

The specific capacities of the NiMoO$_4$ cathode at different current densities were calculated from the galvanostatic discharge curves according to the equation below.

$$C = \frac{I \times \Delta t}{3,600 \times m} \tag{6}$$

where $C$ (mAh g$^{-1}$) is the specific capacity, $I$ (mA) is the discharge current and $m$ (g) is the mass of the active material.

Electrochemical impedance spectroscopy was performed under a sinusoidal signal over a frequency range from 0.01 to $10^5$ Hz with a magnitude of 10 mV. The internal resistance ($R_s$) was also determined from the galvanostatic discharge curves by dividing the voltage drop at the beginning of the discharge ($V_{drop}$) by the applied constant current, $I$ (A) according to the formula below.

$$R_s = \frac{V_{drop}}{2I} \tag{7}$$

**Fabrication and evaluation of supercapacitor devices.** A supercapacitor was fabricated with NiMoO$_4$ nanowires as cathode and FeOOH nanoparticles as anode, which were separated with glass fibre filter paper in 2 M KOH electrolyte. The overall volume of the NiMoO$_4$//FeOOH HSCs includes the active materials, current collectors and separator.

The mass ratio of the positive to negative electrode is obtained by using the equation below.

$$\frac{m_+}{m_-} = \frac{(C_- \cdot \Delta V_-)}{(C)} \tag{8}$$

where $m_+$ and $m_-$ are the mass loading of the NiMoO$_4$ and FeOOH electrodes, respectively, $C_-$ is the specific capacitance of the FeOOH electrode, $C$ is the specific capacity of the NiMoO$_4$ electrode and $\Delta V_-$ is the potential window of the FeOOH electrode.

The constant float voltage method was carried out to test the stability of the NiMoO$_4$//FeOOH HSC using a battery test system (LAND CT2001A). In brief, a constant voltage of 1.7 V was applied to an assembled supercapacitor device with NiMoO$_4$ and FeOOH electrodes and 2 M KOH electrolyte. Three charging/discharging cycles from 0 to 1.7 V were performed at a constant current density of 2.5 A g$^{-1}$ every 10 h to quantify the corresponding retaining specific capacitance. The total test time was 450 h.

The gravimetric energy density and power density of the as-fabricated HSC were calculated based on the formula shown below.

$$E = \frac{\int I \cdot V(t) dt}{3.6m} \tag{9}$$

$$P = \frac{3600E}{\Delta t} \tag{10}$$

where $E$ (Wh kg$^{-1}$) is the energy density, $P$ (W kg$^{-1}$) is the power density, $I$ (A) is the discharge current, $V(t)$ is the discharge voltage excluding the IR drop, $m$ (g) is the total mass of the active material (cathode and anode), d$t$ is the time differential and $\Delta t$ (s) is the discharge time.

The volumetric capacitance, energy density and power density of the NiMoO$_4$//FeOOH HSC were calculated based on the formulas below.

$$C_v = \frac{I \Delta t}{V \Delta V} \tag{11}$$

$$E = \frac{\int I \cdot V(t) dt}{3.6V} \tag{12}$$

$$P = \frac{3600E}{\Delta t} \tag{13}$$

where $C_v$ (F cm$^{-3}$) is the volumetric capacitance, $I$ (A) is the discharge current, $\Delta t$ (s) is the galvanostatic discharge time, $\Delta V$ is the voltage range excluding the potential drop, $V$ (cm$^3$) is the total volume of the supercapacitor device (includes cathode, anode and separator), $E$ (mWh cm$^{-3}$) is the volumetric energy density of the supercapacitor device, $V(t)$ is the discharge voltage excluding the IR drop and $P$ (mW cm$^{-3}$) is the volumetric power density of the supercapacitor device.

**Data availability.** All relevant data supporting the findings of this study are available on request from the corresponding author.

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

## Acknowledgements

This work was supported by the National Key Research and Development Program of China (2016YFA0202603), the National Basic Research Program of China (2013CB934103), the Programme of Introducing Talents of Discipline to Universities (B17034), the National Natural Science Foundation of China (51521001, 51502226, 21673171), the National Natural Science Fund for Distinguished Young Scholars (51425204), the Fundamental Research Funds for the Central Universities (WUT: 2015-YB-002, 2016III001, 2016III002) and the Students Innovation and Entrepreneurship Training Program (20151049701006). L.B.Q. would like to acknowledge the support from The Monash Centre for Atomically Thin Materials.

## Author contributions

K.A.O. and L.B.Q. contributed equally to this work. L.Q.M., K.A.O. and L.B.Q. conceived and designed the experiments. K.A.O., L.B.Q. and C.Y. carried out most of the experiments and analyzed the data. K.N.Z. carried out the Reitveld XRD characterization. Z.Y.W. and K.A.O. carried out the XPS characterization. K.A.O., L.B.Q., L.Z. and L.Q.M. cowrote and revised the paper. All authors commented on and discussed the results.

## Additional information

**How to cite this article**: Owusu, K. A. *et al.* Low-crystalline iron oxide hydroxide nanoparticle anode for high performance supercapacitors. *Nat. Commun.* **8**, 14264 doi: 10.1038/ncomms14264 (2017).

**Publisher's note**: 

