## [Peer Review File · Nature Communications]

Reviewers' comments:

Reviewer #1 (Remarks to the Author):

A. Summary of the key results.

The authors present an aqueous asymmetric supercapacitor cell using low-crystalline FeO(OH) as a negative electrode material and working at 1.7 V with. The main claim is the maximum energy density of the asymmetric supercapacitor cell (104 Wh kg⁻¹) and the highest capacitance of FeO(OH) (1066 F/g). Those are one of the highest numbers for aqueous systems. The energy density is comparable to that of hybrid organic-based cells using a carbon anode.

B. Originality and interest.

This work claims the highest capacitance among negative FeO(OH) electrodes, although slightly lower values were recently published (for example, supporting ref. 7).

C. Data & methodology.

The data and methodology are stated clearly. However, it is not certain that the improvements coming from FeO(OH) can be made use of in real supercapacitors. As this paper aims at introducing a viable full cell configuration, all practical requirements must be satisfied.

A number of technical clarifications are needed to make sure the performance of the proposed asymmetric configuration can be translated to a practical device.

D. Appropriate use of statistics.

Statistic data need to be stated explicitly. How many electrodes of each type were tested? What was the variation in performance for measurements with multiple sets of electrodes? What is the variation in the mass loading and thickness of the electrodes?

E. Conclusions.

The conclusions are stated correctly, but need to be validated by additional data/experiments as detailed below.

F. Suggested improvements: experiments, data for possible revision

1. The main point needing clarification is the mass loading (1.6 mg/cm^2 for FeO(OH)) as it is quite different from that typical of industrial porous carbon electrodes (about 10 mg/cm^2 , for example, electrodes from WL Gore & Associates). On one hand, as the gravimetric capacitance of activated carbon electrodes is usually comprised between 100 and 200 F/g , the capacitance (F) of an activated carbon electrode with a mass loading of 10 mg/cm^2 can be quite comparable to that of a FeO(OH) electrode with a mass loading of 1.6 mg cm^{-2} and a capacitance of about 1000 F/g . On the other hand, a loading of 10 mg/cm^2 of active carbon layer make other cell components (current collectors, separators, electrolyte, etc.) contribute less to the total mass of a packaged device than do 1.5 mg/cm^2 of FeO(OH) . Therefore, practical improvements in the gravimetric capacitance, energy density, etc. may not be substantial (Science, 334 (2011) 917).

Therefore, fair comparison with activated carbon electrodes requires using a similar mass loading. If the superior gravimetric capacitance of FeO(OH) is maintained with an electrode mass typical of commercial-grade electrodes, a rigorous proof will be provided for the higher energy density of asymmetric cells using low-crystalline FeO(OH) instead of carbons. An estimation of energy and power density of packaged devices needs to be done (the present submission provides calculations on the materials basis only).

2. The only values provided for mass loading are 1.5 and 1.6 mg cm^{-2} for NiMoO_4 and FeO(OH) (page 10, line 295). Are they used in 3-electrode cells for testing the anode and cathode materials or are they adjusted to the asymmetric cell according to equation 2? What is the mass loading of each material used in 3-electrode measurements and in the asymmetric cell? How does it affect the capacitance and rate capability? What is the mass balance calculated according to Equation 2? What is the specific current density chosen for mass-balancing the asymmetric cell?

3. The more correct stability test for supercapacitors is float voltage rather than cyclability (JPowSources, 225 (2013), 84). This test induces cell failure in a more reasonable time (capacitance retention over 10000 cycles is commonly reported, but is below the requirements for real cells). Moreover, float voltage corresponds to real usage conditions, and would be especially useful as the full asymmetric cell is claimed to work at 1.7 V in 2M KOH electrolyte, which is rather uncommon. It is highly recommended to include the floating test. It would also be useful to specify the working potential windows of the separate electrodes in the asymmetric cell with respect to the water oxidation and reduction potentials of 2M KOH solution.

4. Volumetric values of capacitance, energy, power are more important to practice, but they are missing. Volumetric values can provide the most important added value and need to be included to the manuscript. Volumetric energy density is enhanced by using high-density pseudocapacitive oxides instead of carbons (Goubard-Bretesché et al, Electrochim.Acta, 2016). The thickness of electrodes also needs to be specified.

5. NiMoO₄ works as a typical battery electrode with the formal capacitance highly dependent on the selected potential range (this is reflected in the redox peaks of cyclic voltammograms and the galvanostatic discharge plateau in Fig. S6a). Therefore, it does not seem to be relevant to treat this material as a pseudocapacitive electrode (Science, 343 (2014), 1210; J. Electrochem. Soc., 162 (2015), A5185). Capacity needs to be calculated instead of capacitance for NiMoO₄ throughout the text and in Figures 5(b,c) and S6.

Minor technical remarks:

1. In discussing the performance of electrode materials, it is more correct to refer to the potential (not voltage) window vs SCE (Formula 2).
2. Fig. S1. does not show any electrochemical signature in the indicated potential range (as referred to in the text, page 4, line 95), it is just a schematic picture of different synthesis steps.
3. Fig. S2, S5 "Surface area and pore size distribution of" There is nothing about surface area in both graphs. Isotherms are shown instead of surface area.
4. Fig. S3 Caption. FeOOH is termed "cathode" although throughout the main text it is correctly referred to as "anode".

G. References: appropriate credit to previous work?

Yes.

H. Clarity and context: lucidity of abstract/summary, appropriateness of abstract, introduction and conclusions

The paper is written clearly and is easy to follow. The context is also clearly defined and addresses an important challenge: increasing the energy density of supercapacitors.

Reviewer #2 (Remarks to the Author):

Owusu et al. reported an interesting method to fabricate low-crystalline FeOOH nanoparticle-decorated carbon fiber cloth (CFC) and demonstrated the supercapacitor application of such material. The key contribution is the achievement of excellent cycling stability through creating a crystallinity phase of the active material, i.e. FeOOH. While the method described here is useful, the ideas of (1) using Fe- and Ni-based electrode materials to assemble asymmetric supercapacitors and (2) using amorphous-like electrode materials to improve cycling stability have been demonstrated in other studies already. Therefore this work as a follow-up study is more suitable to be submitted to other specialized journals after addressing the following points:

1. The Rietveld refinement results (R_{wp} , R_p , χ^2) should be provided.
2. Fig. 6a and 6b seem to suggest that water splitting contributes significantly to the total capacitance at high operation voltage. A quantitative analysis on the contribution of water splitting is recommended.
3. The bulk resistances were estimated from the EIS measurements. The values estimated from the galvanostatic charge/discharge curves should also be provided for comparison.

Reviewer #3 (Remarks to the Author):

Comments to NCOMMS-16-09875

The manuscript entitled "Low-crystalline FeOOH nanoparticle anode with high comprehensive electrochemical performance for advanced asymmetric supercapacitors" reports the synthesis of low-crystalline FeOOH nanoparticle anode and its high comprehensive performance for advanced asymmetric supercapacitors when assembled with NiMoO₄ nanowire cathode. The corresponding analyses, such as XRD, XPS, SEM, TEM, BET, CV, Charge/discharge curves and Cycle performance are systematically carried out in the manuscript. Due to the high comprehensive performance of the low crystalline FeOOH nanoparticles, the assembled NiMoO₄//FeOOH ASC device displays good electrochemical performance at a wide potential window. I think this work is well down with certain originality. But there are some modifications should be made.

1. The expression of the abstract part should be improved in order to further highlight the manuscript.

2. The morphology of the obtained FeOOH nanoparticles should be further illustrated by complementing detailed SEM images with different magnification.

3. Why did the authors choose NiMoO₄ nanowires to be the cathode among numerous alternatives? I think some explanations on it should be given in the manuscript.

4. As mentioned in the manuscript by the authors, asymmetric supercapacitors have been extensively studied, and FeOOH-based anodes for asymmetric supercapacitors also have been studied already. Meanwhile, as far as I know, the nanoparticle morphology is not novel enough, the NiMoO₄ cathode used in supercapacitors is not innovative, and the free-standing electrode designing with the direct growth of active materials on conductive and porous substrates has been investigated extensively. Thus, what are the innovation points of this manuscript? And the intrinsic reasons for the superior electrochemical performance of the NiMoO₄//FeOOH ASC should be further illustrated.

Reviewers' comments:

Response to Reviewer #1:

We would like to thank Reviewer #1 for the deep review and kind comments about our manuscript. We welcome the opportunity to address and clarify the issues raised in the reviewer's comments and we are optimistic that the additional experiments and revisions carried out to address the reviewer's comments substantively strengthen our revised manuscript. Our responses to the points raised in the reports are as follows:

Reviewer #1 (Remarks to the Author):

A. Summary of the key results.

The authors present an aqueous asymmetric supercapacitor cell using low-crystalline FeO(OH) as a negative electrode material and working at 1.7 V with. The main claim is the maximum energy density of the asymmetric supercapacitor cell (104 Wh kg⁻¹) and the highest capacitance of FeO(OH)(1066 F/g). Those are one of the highest numbers for aqueous systems. The energy density is comparable to that of hybrid organic-based cells using a carbon anode.

B. Originality and interest.

This work claims the highest capacitance among negative FeO(OH) electrodes, although slightly lower values were recently published (for example, supporting ref. 7).

C. Data & methodology.

The data and methodology are stated clearly. However, it is not certain that the improvements coming from FeO(OH) can be made use of in real supercapacitors. As this paper aims at introducing a viable full cell configuration, all practical requirements must be satisfied.

A number of technical clarifications are needed to make sure the performance of the proposed asymmetric configuration can be translated to a practical device.

D. Appropriate use of statistics.

Statistic data need to be stated explicitly. How many electrodes of each type were tested? What was the variation in performance for measurements with multiple sets of electrodes? What is the variation in the mass loading and thickness of the electrodes?

Our Response: We thank the reviewer for the valuable suggestions and comments. According to the Reviewer #1's suggestions, more than ten samples of the low-crystalline FeOOH nanoparticle anode were tested. The mass loadings of the FeOOH anode are 1.4 – 2 mg cm⁻², the thickness of the electrodes is ~0.35 mm including the current collector. The FeOOH anode exhibits specific capacitances ranging from 998 – 1092 F g⁻¹ at 1 A g⁻¹. The performance distribution of selected FeOOH electrodes with different mass loadings and thickness are shown below (Supplementary Figure 6a).

Supplementary Figure 6a | Electrochemical performance distribution of selected FeOOH nanoparticle electrodes.

The following descriptions have also been added in the supplementary information of the revised manuscript: "As depicted in Supplementary Fig. 6a, the low-crystalline FeOOH nanoparticle anode exhibits specific gravimetric capacitances ranging from 998 – 1092 F g⁻¹ at 1 A g⁻¹ when the mass loading is between 1.4 – 2 mg with an electrode thickness of ~0.35mm (including the current collector)."

E. Conclusions.

The conclusions are stated correctly, but need to be validated by additional data/experiments as detailed below.

F. Suggested improvements: experiments, data for possible revision

1. The main point needing clarification is the mass loading (1.6 mg/cm² for FeO(OH)) as it is quite different from that typical of industrial porous carbon electrodes (about 10 mg/cm², for example, electrodes from WL Gore & Associates). On one hand, as the gravimetric capacitance of activated carbon electrodes is usually comprised between 100 and 200 F/g, the capacitance (F) of an activated carbon electrode with a mass loading of 10 mg/cm² can be quite comparable to that of a FeO(OH) electrode with a mass loading of 1.6 mg cm⁻² and a capacitance of about 1000 F/g. On the other hand, a loading of 10 mg/cm² of active carbon layer make other cell components (current collectors, separators, electrolyte, etc.) contribute less to the total mass of a packaged device than do 1.5 mg/cm² of FeO(OH). Therefore, practical improvements in the gravimetric capacitance, energy density, etc. may not be substantial (Science, 334 (2011) 917).

Therefore, fair comparison with activated carbon electrodes requires using a similar mass loading. If the superior gravimetric capacitance of FeO(OH) is maintained with an electrode mass typical of commercial-grade electrodes, a rigorous proof will be provided for the higher energy density of asymmetric cells using low-crystalline FeO(OH) instead of carbons. An estimation of energy and power density of packaged devices needs to be done (the present submission provides calculations on the materials basis only).

Our Response: We are thankful to the reviewer for the very helpful suggestions. We agree with reviewer #1 that high mass loading in supercapacitor electrodes is an important performance metric as it decreases the effect of the other device components (separator, current collectors, etc.) on the performance of the full device. According to reviewer #1's suggestions, we studied the electrochemical performances of the FeOOH anode with different mass loadings (1.6, 3.0, 5.6, and 9.1 mg cm⁻²). At 1 A g⁻¹, the FeOOH anodes display specific capacitances of 1066, 966, 827 and 716 F g⁻¹, with mass loadings of 1.6, 3.0, 5.6 and 9.1 mg cm⁻² respectively (Figure 3c). A mass loading of 9.1 mg cm⁻² is comparable to the mass loading of typical industrial porous carbon electrodes (10 mg cm⁻²). At such a high mass loading, the specific, area, and volumetric capacitances reach 716 F g⁻¹, 6.5 F cm⁻² (Figure 3c), and 186 F cm⁻³ (Figure 3d), respectively. These values are much higher than typical industrial porous carbon electrodes.

Figure 3c | Specific gravimetric and area capacitances of the FeOOH nanoparticle anode at different mass loadings.

Figure 3d | Volumetric capacitance of the FeOOH nanoparticle anode at different mass loadings.

At a high mass loading of 9.1 mg cm^{-2} , the FeOOH anode retains $\sim 60\%$ of the initial capacitance at 20 A g^{-1} (427 F g^{-1}), demonstrating its high rate capability (**Supplementary Figure 6b**). In addition, the FeOOH anode with a high mass loadings of 9.1 mg cm^{-2} also shows excellent cycling stability with 86% of the initial capacitance retained after 10000 cycles at 15 A g^{-1} (**Figure 3e**).

Supplementary Figure 6b | Rate capability of the low-crystalline FeOOH nanoparticle anode at different mass loadings.

Figure 3e | Cycling performance of the FeOOH nanoparticle anode at 1.6 and 9.1 mg cm⁻².

A NiMoO₄/FeOOH packaged device with a total active material mass loading of ~2.8 mg presented in the original manuscript presents a maximum gravimetric energy density of ~7 Wh kg⁻¹ and a maximum power density of 1800 W kg. A low mass loading of ~2.8 mg accounts for only ~ 6.5 wt % of the packaged device. The relatively low energy density of the packaged device is limited by the weight of the current collectors (nickel foam) which doesn't contribute to the energy storage (*J. Electrochem. Soc.* **162**, A5185–A5189 (2015)).

To achieve realistic values, we also estimated the gravimetric and volumetric energy densities of a NiMoO₄/FeOOH packaged device with a total active mass loading of 24.5 mg. The total

weight of the assembled NiMoO₄//FeOOH supercapacitor is ~ 70.2 mg (weight of FeOOH active material + carbon cloth current collector = 20 mg; weight of glass fiber filter separator = 8.2 mg; weight of NiMoO₄ active material + nickel foam current collector = 42 mg). The total weight of the active materials (~ 24.5 mg) accounts for 35 wt % of the total packaged mass. The packaged NiMoO₄//FeOOH hybrid supercapacitor displays a maximum energy density of 31.44 Wh kg⁻¹ at a power density of 305 W kg⁻¹ and a maximum power density of 4976 W kg⁻¹ at an energy density of 12.72 Wh kg⁻¹ (Supplementary Figure 13b). The volumetric capacitance, energy density, and power density of the packaged device are also determined. The total volume of the packaged device is 0.13 cm³ (including the volume of current collectors, separator and active materials). The NiMoO₄//FeOOH packaged device displays a maximum volumetric capacitance of 42.96 F cm⁻³ at 1 A g⁻¹, a maximum volumetric energy density of 17.24 Wh L⁻¹ at a volumetric power density of 167.72 W L⁻¹, and a maximum volumetric power density of 2736 W L⁻¹ at a volumetric energy density of 7 Wh L⁻¹ (Figure 6f).

Supplementary Figure 13b | Gravimetric energy and power densities of the NiMoO₄//FeOOH packaged device. Active electrode material accounts for 35% of the total weight.

Figure 6f | Volumetric energy and power density of the NiMoO₄//FeOOH packaged device. Active material mass accounts for 35 % of the total packaged weight.

The following descriptions have also been added in the revised manuscript:

"With mass loadings of 1.6, 3.0, 5.6 and 9.1 mg cm⁻², the low-crystalline FeOOH nanoparticle anode displays specific gravimetric capacitances of 1066, 996, 827 and 716 F g⁻¹ at 1 A g⁻¹, respectively (Fig. 3c). The areal and volumetric capacitances of the FeOOH anode with a high mass loading of 9.1 mg cm⁻² can reach as high as 6.5 F cm⁻² (Fig. 3c) and 186 F cm⁻³ (Fig. 3d)."

"For the FeOOH anode with a mass loading of 1.6 mg cm⁻², 91 % of the initial capacitance can be retained after 10000 charge/discharge cycles at 30 A g⁻¹, while 86 % of the initial capacitance is retained for the anode with a mass loading of 9.1 mg cm⁻² after 10000 cycles at 15 A g⁻¹."

"At a high mass loading of 9.1 mg cm⁻², the FeOOH anode retains ~60 % of the initial capacitance at 20 A g⁻¹ (1 A g⁻¹ = 716 F g⁻¹; 20 A g⁻¹ = 427 F g⁻¹) (Supplementary Fig. 6b). 67 % of the capacitance is retained in a 1 – 30 A g⁻¹ current density range (1 A g⁻¹ = 827 F g⁻¹; 20 A g⁻¹ = 555 F g⁻¹) at a mass loading of 5.6 mg cm⁻²."

"For practical applications, a NiMoO₄/FeOOH packaged device with active materials accounting for 35 % of the total weight is also assembled. It displays a volumetric capacitance of 42.96 F cm⁻³, a maximum energy density of 31.44 Wh kg⁻¹ at a power density of 305 W kg⁻¹, and a maximum power density of 4976 W kg⁻¹ at an energy density of 12.72 W kg⁻¹ (Supplementary Fig. 13). Lastly, the packaged device displays maximum volumetric energy and power densities of 17.24 Wh L⁻¹ and 2736.08 W L⁻¹, respectively (Fig. 6f)."

2. The only values provided for mass loading are 1.5 and 1.6 mg cm⁻² for NiMoO₄ and FeO(OH) (page 10, line 295). Are they used in 3-electrode cells for testing the anode and cathode materials or are they adjusted to the asymmetric cell according to equation 2? What is the mass loading of each material used in 3-electrode measurements and in the asymmetric cell? How does it affect the capacitance and rate capability? What is the mass balance calculated according to Equation 2? What is the specific current density chosen for mass-balancing the asymmetric cell?

Our Response: We thank the reviewer for the very valuable comments. In our original manuscript, the mass loadings of the cathode and anode for electrochemical performance tests in the 3-electrode cells are 1.5 and 1.6 mg cm⁻², respectively. After considering the reviewer's suggestions carefully, we further increase of the mass loading of the anode and cathode materials in the revised manuscript. The mass loading of the FeOOH anode can be tuned from 1.4 to 9.1 mg cm⁻², while the mass loading the NiMoO₄ cathode can be tuned from 1.5 to 8.0 mg cm⁻². The electrochemical performances of the FeOOH anode with different mass loadings are provided in **Figure 3b and 3e** of the revised manuscript. Generally, the specific capacitance decreases with the increase of mass loading.

For the fabrication of the asymmetric cells, the mass loadings of the cathode and anode electrodes were adjusted according to the charge-balance equation shown in equation 2. The specific current density we choose for mass-balancing the asymmetric cells is 5.5 A g⁻¹. At such a current density, the NiMoO₄ cathode delivers a specific capacitance of 1347 F g⁻¹ and the FeOOH anode delivers a specific capacitance of 892 F g⁻¹ (**Supplementary Figure 9**). The potential windows for the cathode and anode are 0.5 and 1.2 V, respectively. The mass balance was determined as follows.

$$\frac{m_+}{m_-} = \frac{C_- \cdot \Delta V_-}{C_+ \cdot \Delta V_+} = \frac{(892 \times 1.2)}{(1347 \times 0.5)}$$

$$\frac{m_+}{m_-} = 1.59$$

Based on this mass ratio, two asymmetric cells were assembled. The first asymmetric cell has an active material mass, cathode mass, and anode mass of 2.8, 1.7, and 1.1 mg, respectively. And mass ratio is 1.55 for the first asymmetric cell. The second asymmetric cell has an active material mass, cathode mass, and anode mass of 24.5, 15, and 9.5 mg, respectively. The mass ratio is 1.58 for the second asymmetric cell.

Supplementary Figure 9 | Galvanostatic discharge curves of FeOOH and NiMoO₄ at 5.5 A g⁻¹ for mass balancing. (a) Galvanostatic discharge curve of the low-crystalline FeOOH nanoparticle anode at 5.5 A g⁻¹. (b) Galvanostatic discharge curve of the NiMoO₄ nanowire cathode.

The following descriptions have also been added in the revised manuscript:

"The NiMoO₄ cathode and FeOOH anode are mass balanced at 5.5 A g⁻¹ (Supplementary Fig. 9) and the optimal mass ratio is calculated to be 1.59."

The following descriptions have also been added in Supplementary Information of the revised manuscript:

The specific current density we choose for mass-balancing the asymmetric cells is 5.5 A g^{-1} . At such a current density, the NiMoO_4 cathode delivers a specific capacitance of 1347 F g^{-1} and the FeOOH anode delivers a specific capacitance of 892 F g^{-1} (Supplementary Figure 9). The potential windows for the cathode and anode are 0.5 and 1.2 V, respectively. The mass balance was determined as follows.

$$\frac{m_+}{m_-} = \frac{C_- \cdot \Delta V_-}{C_+ \cdot \Delta V_+} = \frac{(892 \times 1.2)}{(1347 \times 0.5)}$$
$$\frac{m_+}{m_-} = 1.59$$

3. The more correct stability test for supercapacitors is float voltage rather than cyclability (JPowSources, 225 (2013), 84). This test induces cell failure in a more reasonable time (capacitance retention over 10000 cycles is commonly reported, but is below the requirements for real cells). Moreover, float voltage corresponds to real usage conditions, and would be especially useful as the full asymmetric cell is claimed to work at 1.7 V in 2 M KOH electrolyte, which is rather uncommon. It is highly recommended to include the floating test. It would also be useful to specify the working potential windows of the separate electrodes in the asymmetric cell with respect to the water oxidation and reduction potentials of 2 M KOH solution.

Our Response: We thank the reviewer for his/her valuable suggestions. We agree with the reviewer that the float voltage test is a more correct stability test for supercapacitors (*Science* **341**, 534–537 (2013); *J. Power Sources* **225**, 84–88 (2013)). According to the reviewer's suggestions, we carried out the float voltage test and the results were provided in **Figure 6d** of the revised manuscript. .

Figure 6d | Float voltage stability test of the NiMoO₄/FeOOH HSC for 450 hours.

In the float voltage test, a voltage of 1.7 V was applied to an assembled supercapacitor device with NiMoO₄ cathode, FeOOH anode, and KOH electrolyte (2.0 M). Three charge and discharge cycles were performed with a constant current density of 2.5 A g⁻¹ every 10 hours to study the stability of the supercapacitors. The NiMoO₄/FeOOH hybrid supercapacitor displays exceptional stability during a long test time of 450 hours, with no loss in specific capacitance. The float voltage test confirms that the NiMoO₄/FeOOH hybrid supercapacitor is stable and functional in the wide potential window of 1.7 V in 2.0 M KOH electrolyte (Figure 6d).

The following description has been provided in the revised manuscript: "The float voltage test, a more demanding test than the conventional charge/discharge cycling was also used to study the stability of the NiMoO₄/FeOOH HSC^{57,58}. For a test time of 450 hours, the NiMoO₄/FeOOH hybrid supercapacitor displays exceptional stability with no loss in capacitance (Fig. 6d)."

"57. Weingarh, D., Foelske-Schmitz, A. & Kötz, R. Cycle versus voltage hold: Which is the better stability test for electrochemical double layer capacitors? *J. Power Sources* **225**, 84–88 (2013)."

"58. Yang, X., Cheng, C., Wang, Y., Qiu, L. & Li, D. Liquid-mediated dense integration of graphene materials for compact capacitive energy storage. *Science* **341**, 534–537 (2013)."

Furthermore, based on the reviewer's useful suggestions, we identified the working potentials of the NiMoO₄ and FeOOH electrodes with respect to the water oxidation and reduction potentials in 2 M KOH electrolyte through cyclic voltammetry (CV), galvanostatic charge/discharge, and linear sweep voltammetry (LSV) tests.

Supplementary Figure 10 | Water oxidation and reduction potentials of the FeOOH and NiMoO₄ electrodes in 2 M KOH electrolyte. (a) Discharge curve of low-crystalline FeOOH nanoparticles at the current density of 1.2 A g⁻¹ in 2 M KOH electrolyte. **(b)** CV curve of low-crystalline FeOOH nanoparticles at the scan-rate of 1 mV s⁻¹ in 2 M KOH electrolyte. **(c)** Hydrogen reduction potential of the low-crystalline FeOOH nanoparticles in 2 M KOH electrolyte. **(d)** Charge curve of the NiMoO₄ nanowires at the current density of 1 mA cm⁻² in 2 M KOH electrolyte. **(e)** CV curve of the NiMoO₄ nanowires at a scan rate of 1 mV s⁻¹ in 2 M KOH electrolyte. **(f)** Water oxidation potential of the NiMoO₄ nanowires at the scan rate of 1 mV s⁻¹ in 2 M KOH electrolyte.

As evidenced in Supplementary Figure 10a, 10b, and 10c, the low-crystalline FeOOH nanoparticle anode exhibits good stability in the potential range of -1.2 to 0 vs. SCE. Meanwhile, the reduction for the hydrogen evolution occurs at around -1.25 vs. SCE. Likewise, the NiMoO₄ nanowire cathode shows good stability in 2.0 M KOH electrolyte up to 0.52 V vs. SCE after which the water starts to decompose and the emergence of oxygen is observed (Supplementary Figure 10d, 10e and 10f). In summary, there is no significant contribution from water splitting at an electrochemical window of 1.7 V. However, if the potential window is extended to 1.9 V, O₂ evolution is observed and water splitting contributes significantly to the capacitance (Figure 6a).

The following descriptions have also been added in the revised manuscript:

"The optimal potential window of the assembled HSC is determined to be 1.7 V. This is in good agreement with the working potential windows of the separate electrodes with respect to the water oxidation and reduction potentials in 2 M KOH electrolyte (Supplementary Fig. 10)."

4. Volumetric values of capacitance, energy, power are more important to practice, but they are missing. Volumetric values can provide the most important added value and need to be included to the manuscript. Volumetric energy density is enhanced by using high-density pseudocapacitive oxides instead of carbons (Goubard-Bretesché et al, *Electrochim.Acta*, 2016). The thickness of electrodes also needs to be specified.

Our Response: We thank the reviewer for the very valuable suggestion. According to the reviewer's suggestions, the volumetric capacitance (Figure 6c), energy density, and power density (Supplementary Figure 12) are provided in the revised manuscript.

The hybrid NiMoO₄/FeOOH supercapacitor was assembled by sandwiching a glass fiber filter (thickness = 0.2 mm) separator between the NiMoO₄ cathode (thickness = 0.4 mm) and FeOOH anode (thickness = 0.35 mm). The total volume of the NiMoO₄ cathode, FeOOH anode, and glass fiber separator is 0.0895 cm³. The volumetric capacitance, energy density, and power density of the as-fabricated NiMoO₄/FeOOH packaged device are calculated according to the equations shown below.

$$C_v = \frac{I\Delta t}{V\Delta V} \quad (1)$$

$$E = \frac{\int I \cdot V(t) dt}{3.6V} \quad (2)$$

$$P = \frac{3600E}{\Delta t} \quad (3)$$

where C_v ($F \text{ cm}^{-3}$) is the volumetric capacitance, I (A) is the discharge current, Δt (s) is the galvanostatic discharge time, ΔV is the voltage range excluding the potential drop, V is the total volume of the cathode, anode, and separator, E (mWh cm^{-3}) is the volumetric energy density of the supercapacitor device, $V(t)$ is the discharge voltage excluding the IR drop and P (mW cm^{-3}) is the volumetric power density of the supercapacitor device.

Figure 6c. | The specific gravimetric and volumetric capacitances of the HSC at different current densities.

Supplementary Figure 12 | Volumetric energy density and power density of the NiMoO₄//FeOOH packaged device. Active electrode materials account for 6.5 % of the total weight.

The NiMoO₄//FeOOH hybrid supercapacitor displays very high volumetric capacitances; 8.24 and 5.53 F cm⁻³ at 1.5 and 22.5 A g⁻¹, respectively (Figure 6c). The volumetric energy densities are 3.15, 2.94, 2.8, 2.3, 1.89, 1.50 and 0.68 mWh cm⁻³ at power densities of 38.33, 78.99, 98.54, 181.2, 247.4, 303.4 and 330.6 mW cm⁻³, respectively (Supplementary Figure 12).

A packaged device with an active material mass loading of 24.5 mg, which accounts for 35 % of the total package weight, delivers volumetric capacitances of 42.96 and 17.42 F cm⁻³ at 1 and 10 A g⁻¹, respectively (Supplementary Figure 13a). The volumetric energy densities are 17.24, 13.70, 11.78, 10.21, 8.22 and 7.00 Wh L⁻¹ at volumetric power densities of 167.72, 351.60, 555.56, 921.13, 1634.41 and 2736.08 W L⁻¹ (Figure 6f).

Supplementary Figure 13a | Volumetric capacitance as a function of current density of the NiMoO₄//FeOOH packaged device. Active electrode material accounts for 35% of the total weight.

The following descriptions have also been added in the revised manuscript:

"Volumetric capacitance, energy density and power density are very important parameters for practical applications of supercapacitors⁴⁹. The NiMoO₄//FeOOH packaged device displays high volumetric capacitances; even though the active material mass accounts for just 6.5 wt% of the

packaged device, the volumetric capacitances still reach 8.24 and 5.53 F cm⁻³ at 1.5 and 22.5 A g⁻¹, respectively (Fig. 6c). The HSC device also displays a maximum volumetric energy density of 3.15 mWh cm⁻³ at a power density of 38.33 mW cm⁻³ and a maximum volumetric power density of 330.62 mw cm⁻³ at a energy density of 0.68 mWh cm⁻³ (Supplementary Fig. 12). For practical applications, a NiMoO₄/FeOOH packaged device with active materials accounting for 35 % of the total weight is also assembled. It displays a volumetric capacitance of 42.96 F cm⁻³, a maximum energy density of 31.44 Wh kg⁻¹ at a power density of 305 W kg⁻¹, and a maximum power density of 4976 W kg⁻¹ at an energy density of 12.72 W kg⁻¹ (Supplementary Fig. 13)."

"49. Gogotsi, Y & Simon, P. True performance metrics in electrochemical energy storage. *Science* **334**, 917–918 (2011)."

5. NiMoO₄ works as a typical battery electrode with the formal capacitance highly dependent on the selected potential range (this is reflected in the redox peaks of cyclic voltammograms and the galvanostatic discharge plateau in Fig. S6a). Therefore, it does not seem to be relevant to treat this material as a pseudocapacitive electrode (Science, 343 (2014), 1210; J. Electrochem. Soc., 162 (2015), A5185). Capacity needs to be calculated instead of capacitance for NiMoO₄ throughout the text and in Figures 5(b,c) and S6.

Our Response: We very much appreciate the reviewer's valuable suggestion. According to the reviewer's suggestion, specific capacity (mAh g⁻¹) instead of specific capacitance (F g⁻¹) is used to quantify the electrochemical performance of the NiMoO₄ cathode in the revised manuscript. The specific capacities were calculated from the galvanostatic discharge curves using the equation specified below.

$$C_{specific} = \frac{Q}{M} = \frac{I \times \frac{\Delta t}{3600}}{M} = \frac{I \times \Delta t}{3600M} \quad (4)$$

where C (mAh g⁻¹) is specific capacity, Q is the quantity of charge, I (mA) represents the discharge current, M (g) and Δt (s) stand for the mass of the active material and discharge time respectively.

As shown in **Figure 5b**, the NiMoO₄ electrode delivers a high specific capacity of 223 mAh g⁻¹ (0.33 mAh cm⁻²) at 1 A g⁻¹ and 59 % of the specific capacity can be retained at 30 A g⁻¹ (130 mAh g⁻¹, 0.2 mAh cm⁻²). The long-term cycling performance of the NiMoO₄ nanowires was also studied. The NiMoO₄ nanowires display a specific capacity of 111 mAh g⁻¹ with a retention of 85.1 % after 10000 charge/discharge cycles at 30 A g⁻¹ (**Figure 5c**).

Finally, based on the definition of hybrid supercapacitor as being comprised of a battery or faradaic type material and a capacitive electrode, we have revised the manuscript and substituted asymmetric supercapacitor with hybrid supercapacitor where applicable.

The following descriptions have also been added in the revised manuscript:

"The charge storage mechanism in NiMoO₄ can be ascribed to Faradaic battery-type mechanism from the sharp peaks of the CV curves^{12,27}."

"The specific capacity instead of specific capacitance of the NiMoO₄ cathode was calculated from the discharge curves (Supplementary Fig. 8a) to give realistic values of the energy storage and release^{12,27}. As shown in Fig. 5b and Supplementary Fig. 8b, the NiMoO₄ electrode delivers a high specific capacity of 223 mAh g⁻¹ (0.33 mAh cm⁻²) at 1 A g⁻¹ and 59 % of the capacity can be retained at 30 A g⁻¹ (130 mAh g⁻¹, 0.2 mAh cm⁻²)."

"27. Brousse, T., Bélanger, D. & Long, J. W. To be or not to be pseudocapacitive? *J. Electrochem. Soc.* **162**, A5185–A5189 (2015)."

Figure 5 | Electrochemical performance of the NiMoO₄ nanowire cathode. (a) CV curves. (b) Specific capacity as a function of current density. (c) Cycling performance at 30 A g⁻¹. (d) Nyquist plot, inset is the magnified view of the Nyquist plot in high-frequency region.

Minor technical remarks:

1. In discussing the performance of electrode materials, it is more correct to refer to the potential (not voltage) window vs SCE (Formula 2).

Our Response: We would like to thank the reviewer for the correction. The term "voltage window" has been substituted with "potential window" in the revised manuscript according to the reviewer's suggestion.

2. Fig. S1. does not show any electrochemical signature in the indicated potential range (as referred to in the text, page 4, line 95), it is just a schematic picture of different synthesis steps.

Our Response: We thank the reviewer for the kind reminder. The revised **Supplementary Figure 1** is provided below, in which the potential range for the electrochemical transformation of Fe_2O_3 to FeOOH has been indicated.

Supplementary Figure 1 | Schematic illustration of the synthesis process of FeOOH nanoparticle anode. FeOOH nanoparticle anode was achieved through a novel two-step process which involves the hydrothermal growth of $\alpha\text{-Fe}_2\text{O}_3$ nanoparticles on carbon fiber cloth (CFC) substrates and subsequent transformation during the electrochemical cycles in the potential range between -1.2 and 0 V *versus* saturated calomel electrode (SCE). The low-crystalline FeOOH nanoparticles remain stable after transformation.

3. Fig. S2, S5 "Surface area and pore size distribution of" There is nothing about surface area in both graphs. Isotherms are shown instead of surface area.

Our Response: We would like to thank the reviewer for the kind corrections. Figure S2 and S5 have been revised accordingly. The revised **Supplementary Figure 2** is shown below.

Supplementary Figure 2 | Nitrogen sorption results. The nitrogen adsorption-desorption isotherms of the (a) α -Fe₂O₃ electrode and (b) NiMoO₄ electrode. The insets show the corresponding pore size distributions.

4. Fig. S3 Caption. FeOOH is termed "cathode" although throughout the main text it is correctly referred to as "anode".

Our Response: We thank the reviewer for careful correction. The caption for Supplementary Figure 4 (old Supplementary Figure 3) has been corrected accordingly.

Supplementary Figure 4 | Electrochemical characterization of FeOOH electrode. (a) The galvanostatic charge/discharge curves of FeOOH anode at current densities ranging from 1 to 30 A g⁻¹. SCE (saturated calomel electrode) is used as the reference electrode. (b) A plot of areal capacitance of the FeOOH anode as a function of current density (1 to 30 A g⁻¹).

G. References: appropriate credit to previous work?

Yes.

H. Clarity and context: lucidity of abstract/summary, appropriateness of abstract, introduction and conclusions

The paper is written clearly and is easy to follow. The context is also clearly defined and addresses an important challenge: increasing the energy density of supercapacitors.

Our Response: We very much appreciate the Reviewer's high evaluation of our manuscript.

Reviewer #2 (Remarks to the Author):

Response to Reviewer #2:

We thank the Reviewer #2 very much for his great time and effort in assessing our manuscript. We have performed a series of new experiments and added discussions to address fully the points raised by the reviewer. We believe the new results, discussions, and revisions significantly strengthened the overall quality of our manuscript. Our point-by-point responses to the reviewer's comments are detailed below.

Owusu et al. reported an interesting method to fabricate low-crystalline FeOOH nanoparticle-decorated carbon fiber cloth (CFC) and demonstrated the supercapacitor application of such material. The key contribution is the achievement of excellent cycling stability through creating a crystallinity phase of the active material, i.e. FeOOH. While the method described here is useful, the ideas of (1) using Fe- and Ni-based electrode materials to assemble asymmetric supercapacitors and (2) using amorphous-like electrode materials to improve cycling stability have been demonstrated in other studies already. Therefore, this work as a follow-up study is more suitable to be submitted to other specialized journals after addressing the following points:

Response to General comments: We thank the reviewer for his critical but very valuable comments.

The novelty of this manuscript lies in the finding that not only the surface but also the bulk of α -Fe₂O₃ nanoparticles can be converted into low-crystalline FeOOH during electrochemical activation, which has rarely been reported. The FeOOH functions as an ideal pseudocapacitive anode with dominant capacitive contribution (Figure 7), which offer high specific capacitance without compromising the charge storage kinetics, even at high mass loadings. Such a behavior is very beneficial to the future application of supercapacitors as most reported pseudocapacitive anodes are faradic controlled and the charge transfer would easily be impeded at high mass loadings.

The following description has been added to the revised manuscript:

"Using *ex situ* XRD, XPS, SEM and TEM tests, it has been unambiguously demonstrated that not only the surface but also the bulk of the α -Fe₂O₃ nanoparticles can be converted into low-crystalline FeOOH during the electrochemical activation process, which has been rarely reported."

"The low-crystalline design of the pseudocapacitive anode with high comprehensive performance largely enhances the energy and power densities of the supercapacitor. Therefore, the well-designed low-crystalline FeOOH materials could be a very suitable supercapacitor anode for future practical applications due to its low cost, easy preparation, environmental benignity and high comprehensive electrochemical performance at a wide potential window."

1. The Rietveld refinement results (Rwp, Rp, χ^2) should be provided.

Response to Comment-1: We are grateful to the reviewer for his kind suggestions. The Rietveld refinement results (Rwp, Rp, χ^2) of the NiMoO₄ and α -Fe₂O₃ are shown in the table below.

Supplementary Table 1 | Rietveld refinement results of NiMoO₄ and α -Fe₂O₃

	a (Å)	b (Å)	c (Å)	e_0	Rwp (%)
NiMoO ₄	9.566(0)	8.734(0)	7.649(0)	0.00640	6.853
α -Fe ₂ O ₃	9.566(0)	9.566(0)	7.649 (0)	0.00370	6.593

2. Fig. 6a and 6b seem to suggest that water splitting contributes significantly to the total capacitance at high operation voltage. A quantitative analysis on the contribution of water splitting is recommended.

Response to Comment-2: We thank the reviewer #2 very much for the valuable suggestion. Based on the suggestion of the reviewer, we have carried out additional experiments (CV, galvanostatic charge/discharge and LSV tests) to specify the working potential windows of the separate electrodes in the hybrid cell with respect to the water oxidation and reduction potentials of 2 M KOH electrolyte solution.

Supplementary Figure 10 | Water oxidation and reduction potentials of the FeOOH and NiMoO₄ electrodes in 2 M KOH electrolyte (a) Discharge curve of low-crystalline FeOOH nanoparticles at the current density of 1.2 A g⁻¹ in 2 M KOH electrolyte. (b) CV curve of low-crystalline FeOOH nanoparticles at the scan-rate of 1 mV s⁻¹ in 2 M KOH electrolyte. (c) Hydrogen reduction potential of the low-crystalline FeOOH nanoparticles in 2 M KOH electrolyte. (d) Charge curve of the NiMoO₄ nanowires at the current density of 1 mA cm⁻² in 2 M KOH electrolyte. (e) CV curve of the NiMoO₄ nanowires at a scan rate of 1 mV s⁻¹ in 2 M KOH electrolyte. (f) Water oxidation potential of the NiMoO₄ nanowires at the scan rate of 1 mV s⁻¹ in 2 M KOH electrolyte.

When the hybrid cells were charged to 1.7 V vs. SCE, the water was stable and we didn't observe the evolution of O₂ during the charge process. As evidenced in Supplementary Figure 10a, 10b, and 10c, the low-crystalline FeOOH anode exhibits good stability in the potential range of -1.2 to 0 vs. SCE. Meanwhile, the reduction for the hydrogen evolution occurs at around -1.25 vs. SCE. Likewise, the NiMoO₄ nanowire cathode shows stability in 2 M KOH electrolyte up to 0.52 V vs. SCE after which the water starts to decompose and the emergence of oxygen is observed (Supplementary Figure 10d, 10e, and 10f). In summary, there is no significant contribution from water splitting at an electrochemical window of 1.7 V. However, if the potential window is extended to 1.9 V, O₂ evolution is observed and water splitting contributes significantly to the capacitance (Figure 6a).

Supplementary Figure 11b displays the galvanostatic charge/discharge curves of the assembled NiMoO₄/FeOOH hybrid supercapacitor at different current densities. The charge/discharge curves are all reversible with no contribution from water splitting at 1.7 V and is consistent with several reported literatures (*Nat. Commun.* **4**, 1894 (2013); *Nano Lett.* **14**, 731–736 (2014); *Adv. Mater.* **27**, 4566–4571 (2015))

Supplementary Figure 11b | Galvanostatic charge-discharge curves of the NiMoO₄/FeOOH hybrid supercapacitor.

The following descriptions have also been added in the revised manuscript:

"The optimal potential window of the assembled HSC is determined to be 1.7 V. This is in good agreement with the working potential windows of the separate electrodes with respect to the water oxidation and reduction potentials in 2 M KOH electrolyte (Supplementary Fig. 10)."

3. The bulk resistances were estimated from the EIS measurements. The values estimated from the galvanostatic charge/discharge curves should also be provided for comparison.

Response to Comment-3: We thank the reviewer for the comment and valuable suggestion. Based on the reviewer's suggestions, bulk resistance (R_s) of the individual electrodes was estimated from the voltage drop at the beginning of the galvanostatic discharge curves according to the equation shown below.

$$R_s = \frac{V_{drop}}{2I} \quad (5)$$

Where R_s (Ω) is the bulk resistance, V_{drop} is the voltage drop at the beginning of the galvanostatic discharge curves and I (A) is the discharge current.

A comparison of the bulk resistances of the electrodes estimated from the galvanostatic discharge curves and the Nyquist plots are shown in Table R2.

Supplementary Table 2 | Resistance values of the NiMoO₄ and FeOOH electrodes.

Sample	R_s (Ω , From EIS Simulation)	R_s (Ω , From Potential Drop)	R_{ct} (Ω)
FeOOH	3.59	3.45	0.59
FeOOH (after cycling)	4.10	4.51	0.50
NiMoO ₄	0.72	0.85	0.15

At 1 A g⁻¹, the FeOOH electrode displays a very low voltage drop of 0.0097 V, suggesting a low bulk resistance (R_s) of the electrode (3.45 Ω). Comparison of the R_s value of the FeOOH anode estimated from the voltage drop to the corresponding values obtained from simulation of the EIS measurements shows a minor variation, 3.45 vs. 3.59 Ω (Supplementary Table 2). For the NiMoO₄ nanowire cathode, the bulk resistances estimated from the EIS measurements and the voltage drops at the beginning of the galvanostatic discharge curves are 0.72 and 0.85 Ω , respectively.

Finally, a plot of the voltage drops *versus* current density of the FeOOH and NiMoO₄ electrodes (Supplementary Figure 14) displays a very gentle slope which can be ascribed to the low internal resistance and excellent conductivity of both electrodes.

Supplementary Figure 14 | Voltage drops of the NiMoO₄ and FeOOH electrodes as a function of the current density. Inset shows the equation of the fitting lines.

The following descriptions have also been added in the revised manuscript:

"At 1 A g⁻¹, the FeOOH electrode displays a very low voltage drop of 0.0097 V, suggesting a low internal resistance (R_s) of the electrode (3.45 Ω)⁵⁰."

"50. Zhu, Y. et al. Carbon-based supercapacitors produced by activation of graphene. Science 332, 1537–1541 (2011)."

"A plot of the voltage drops versus current density of both electrodes displays very gentle slopes, which can be ascribed to the low internal resistances and excellent conductivities of the electrodes (Supplementary Fig. 14)."

Reviewer #3 (Remarks to the Author):

Our Response to reviewer 3: We thank the reviewer 3 for the very positive assessment of our work and welcome the opportunity to address and clarify the issues suggested by reviewer 3. Our responses to reviewer 3's comments are detailed below.

Comments to NCOMMS-16-09875

The manuscript entitled "Low-crystalline FeOOH nanoparticle anode with high comprehensive electrochemical performance for advanced asymmetric supercapacitors" reports the synthesis of low-crystalline FeOOH nanoparticle anode and its high comprehensive performance for advanced asymmetric supercapacitors when assembled with NiMoO₄ nanowire cathode. The corresponding analyses, such as XRD, XPS, SEM, TEM, BET, CV, Charge/discharge curves and Cycle performance are systematically carried out in the manuscript. Due to the high comprehensive performance of the low crystalline FeOOH nanoparticles, the assembled NiMoO₄//FeOOH ASC device displays good electrochemical performance at a wide potential window. I think this work is well down with certain originality. But there are some modifications should be made.

1. The expression of the abstract part should be improved in order to further highlight the manuscript.

Response to Comment-1: Thank you reviewer 1 for the very helpful suggestion. The abstract has been polished according to the reviewer's suggestion. The revised Abstract is provided below.

Carbon materials are generally preferred as anodes in supercapacitors, however their low capacitance limits the attained energy density of hybrid and asymmetric supercapacitors. Here, we develop a novel strategy to synthesize low-crystalline FeOOH nanoparticle anode with high comprehensive performance at a wide potential window. The FeOOH nanoparticles present high capacitance at both low and high mass loadings (1066 and 716 F g⁻¹ at 1.6 and 9.1 mg cm⁻² respectively), good rate capability (74.6 % capacitance retention at 30 A g⁻¹) and excellent cycling stability (91 % capacitance retention after 10000 cycles). The excellent performance is attributed to the dominant capacitive charge-storage contribution (~90 % at 5 mV s⁻¹). A NiMoO₄//FeOOH hybrid supercapacitor shows exceptional stability during float voltage test for

450 hours and high energy density of 104 Wh kg^{-1} at a power density of 1.27 kW kg^{-1} . The packaged device delivers high gravimetric and volumetric energy densities of 33.14 Wh kg^{-1} and 17.24 Wh L^{-1} , respectively.

2. The morphology of the obtained FeOOH nanoparticles should be further illustrated by complementing detailed SEM images with different magnification.

Response to Comment-2: We thank the reviewer for the valuable suggestion. The SEM images of the FeOOH nanoparticles at different magnifications are shown below.

Supplementary Figure 3 | SEM images of the FeOOH nanoparticles at different magnifications. Scale bars (a) $5 \mu\text{m}$, (b) 500 nm , (c) 500 nm , (d) 500 nm , (e) 200 nm , and (f) 200 nm .

The following description has also been added to the revised manuscript: "SEM images of the transformed FeOOH show that the nanoparticle morphology is well-maintained (Supplementary Fig. 3)."

3. Why did the authors choose NiMoO_4 nanowires to be the cathode among numerous alternatives? I think some explanations on it should be given in the manuscript.

Response to Comment-3: We are very thankful for the reviewer's valuable suggestion. Amongst numerous alternatives, nickel-based metal oxides have been vastly studied for their potential applications as cathode materials in supercapacitors (*Chem. Soc. Rev.* **41**, 797–828, (2012)). For instance, nickel oxides and hydroxides hold great promise as electrode materials in alkaline KOH electrolytes because of its high specific capacitance, low cost, and easy synthesis. However, their main limitations of poor cycle life and high resistivity need to be addressed. Recently, Ni-based mixed metal oxides, especially NiMoO₄, have emerged as preferential cathode materials due to the rich-redox reactions, improved electronic conductivity, and better electrochemical performance compared with single metal oxides (*Adv. Energy Mater.* **5**, 1401172 (2015); *Energy Environ. Sci.* **12**, 3619–3626 (2013); *ACS Appl. Mater. Interfaces* **6**, 5050–5055 (2014); *ACS Appl. Mater. Interfaces* **5**, 12905–12910 (2013)). Also, nanostructured nickel molybdate shows high specific capacitances and excellent cycling performances. The former is arisen from the high electrochemical activity of nickel ion and the later is resulting from the chemical stability of the molybdate ions (*Nano Energy* **8**, 174–182 (2014); *J. Mater. Chem. A* **1**, 1380–1387 (2013); *Nat. Commun.* **2**, 381 (2011)). Based on these factors, NiMoO₄ nanowires were directly grown on conductive nickel foam and utilized as an ideal cathode for the fabrication of a hybrid supercapacitor. The direct growth NiMoO₄ on the conductive substrates keeps the resistance low, which is very important for electrochemical stability. The NiMoO₄ nanowire morphology possesses direct 1D electronic pathways and reduced ion transport lengths, which permit efficient charge transport and thus high-rate performance.

The following description has been added to the Introduction Section of the revised manuscript: "NiMoO₄ is selected as the cathode due to its improved electronic conductivity and rich redox reactions, arising from the high electrochemical activity of Ni^{28,44,45}".

4. As mentioned in the manuscript by the authors, asymmetric supercapacitors have been extensively studied, and FeOOH-based anodes for asymmetric supercapacitors also have been studied already. Meanwhile, as far as I know, the nanoparticle morphology is not novel enough, the NiMoO₄ cathode used in supercapacitors is not innovative, and the free-standing electrode designing with the direct growth of active materials on conductive and porous substrates has been investigated extensively. Thus, what are the innovation points of this manuscript? And the

intrinsic reasons for the superior electrochemical performance of the NiMoO₄//FeOOH ASC should be further illustrated.

Response to Comment-4: We thank the reviewer for his/her very valuable comments.

The novelty of this manuscript lies in the finding that not only the surface but also the bulk of α -Fe₂O₃ nanoparticles can be converted into low-crystalline FeOOH during electrochemical activation, which has rarely been reported. The FeOOH functions as an ideal pseudocapacitive anode with dominant capacitive contribution (Figure 7), which offer high specific capacitance without compromising the charge storage kinetics, even at high mass loadings. Such a behavior is very beneficial to the future application of supercapacitors as most reported pseudocapacitive anodes are faradic controlled and the charge transfer would easily be impeded at high mass loadings.

The superior electrochemical performance of the as-fabricated NiMoO₄//FeOOH hybrid supercapacitor can be attributed to the following intrinsic factors: (1) The dominant capacitive contribution of the low-crystalline FeOOH nanoparticle anode results in high capacitances in a wide potential window, which translates into the high-energy density of the hybrid device. (2) The FeOOH nanoparticles present short ion diffusion paths which are favorable for fast redox reactions, and the low crystalline structure has the self-adaptive strain-relaxation capability during the charge and discharge processes, leading to high stability. (3) The large surface area of the active materials provides more active sites for charge storage. (4) The direct growth of active materials on conductive substrates eliminates the use of a binder, which often inhibits electrode-electrolyte contact areas and increases the overall resistances of the electrodes. (5) The highly conductive and porous substrates (nickel foam and CFC) provide continuous electronic transport and easy accessibility of the electrolytes to the active materials.

Figure 7 | Capacitive and diffusion-controlled contributions to charge storage. Voltammetric responses for low-crystalline FeOOH nanoparticles at scan rates of (a) 1, (b) 2, and (c) 5 mV s⁻¹. The capacitive contribution to the total current is shown by the shaded region. (d) The capacitance-contribution at different scan rates (1, 2 and 5 mV s⁻¹).

The following description has been added to the revised manuscript:

"Using *ex situ* XRD, XPS, SEM and TEM tests, it has been unambiguously demonstrated that not only the surface but also the bulk of the α -Fe₂O₃ nanoparticles can be converted into low-crystalline FeOOH during the electrochemical activation process, which has been rarely reported."

"The low-crystalline design of the pseudocapacitive anode with high comprehensive performance largely enhances the energy and power densities of the supercapacitor. Therefore,

the well-designed low-crystalline FeOOH materials could be a very suitable supercapacitor anode for future practical applications due to its low cost, easy preparation, environmental benignity and high comprehensive electrochemical performance at a wide potential window."

"The dominant capacitive contribution of the low-crystalline FeOOH nanoparticle anode results in high capacitances in a wide potential window, which translates into the high-energy density of the hybrid device."

REVIEWERS' COMMENTS:

Reviewer #1 (Remarks to the Author):

The Authors have substantially revised the manuscript and addressed the comments made to the initial submission. Although the values of energy and power density are lower with realistic electrodes, they still present sensible improvement over analogous asymmetric cells. The stability of hybrid capacitors is also acceptable on the basis of correct stability tests. Therefore, this paper is suitable for publication in Nature Communications. A few points still need clarifications.

i) The NiMoO₄ electrode works in the potential range totally exceeding the theoretical oxygen evolution potential in 2M KOH (~ 0.163 V vs SCE). An explanation for this would be very helpful, especially with a focus on the absence of water splitting at the water/NiMoO₄ interface at potentials comprised between the theoretical oxygen evolution and the electrochemical activity of NiMoO₄. What is the reason for suppressing oxygen evolution? This point does not seem to be addressed in the literature, but would be helpful to better understand asymmetric configurations with NiMoO₄ and assess their viability.

ii) The trend in Fig. 3d (“Volumetric capacitance of the FeOOH nanoparticle anode at different mass loadings”) may look misleading since the volumetric capacitance of materials is known to decrease with the higher mass loading. This is obviously because the Authors calculate the volumetric capacitance of electrodes including the volume of current collectors. Instead, the volumetric capacitance of active material alone can be presented for the better clarity of data presentation since the volumetric capacitance of the total device (including all of its components) is also provided in the manuscript. Then, the common trend of lower volumetric capacitance for thicker electrode materials can be evidenced and will not delude the reader. Alternatively, the Captions of Fig. 3d can be modified to say explicitly that the volumetric capacitance includes the volume of current collectors, but providing the common trend is still the better option.

iii) As pointed out with respect to the initial submission, the positive electrode is not pseudocapacitive. Therefore, any mentioning of its capacitance is not relevant. To be technically rigorous, the Authors need to make corresponding changes throughout the manuscript. For example, formula (4) in the revised manuscript (and the text and formula below the Supplementary Fig. 9) requires corrections in the denominator by substituting $C_{\Delta V}$ by the capacity of the NiMoO₄ cathode. Please also check the remainder of the text to make sure the capacitance of the positive electrode is not mentioned.

iv) To present clear and unambiguous background, the Authors are advised to clarify the introduction by stating explicitly that they restrict their discussion to AQUEOUS hybrid cells employing a battery-type cathode and a PSEUDOCAPACITIVE anode. Otherwise, a few statements can be surprising to the reader familiar with the other types of hybrid configurations. For example, Page 2: i) “The design of ASC and HSCs result in high energy density and improved cyclability due to the contributions from the different charge storage mechanisms in an extended potential window (up to 2 V)^{28,29}”. The maximum voltage window of a lithium-ion capacitor working in organic electrolyte is

about 3.8 V. Cycle life is also typically lower with asymmetric or hybrid systems than with fully symmetric double layer capacitors; ii) "To obtain full supercapacitors with high energy density, new pseudocapacitive anodes need to be explored to replace the low-capacitance carbon materials³³". Hybrid lithium ion capacitors usually employ battery-type anodes having a higher capacity (graphite, non-porous hard carbons, LTO) than that of capacitive cathodes.

Reviewer #2 (Remarks to the Author):

The authors have properly addressed the issues. I would now recommend the publication of this work on Nature Communications.

Reviewers' comments:

Response to Reviewer #1:

We would like to thank Reviewer #1 for the very kind comments and high evaluation of our revised manuscript. We welcome the opportunity to address and clarify the remaining issues raised in the reviewer's comments. Our responses to the points raised in the reports are as follows:

Reviewer #1 (Remarks to author):

The Authors have substantially revised the manuscript and addressed the comments made to the initial submission. Although the values of energy and power density are lower with realistic electrodes, they still present sensible improvement over analogous asymmetric cells. The stability of hybrid capacitors is also acceptable on the basis of correct stability tests. Therefore, this paper is suitable for publication in Nature Communications. A few points still need clarifications.

i) The NiMoO₄ electrode works in the potential range totally exceeding the theoretical oxygen evolution potential in 2M KOH (~ 0.163 V vs SCE). An explanation for this would be very helpful, especially with a focus on the absence of water splitting at the water/NiMoO₄ interface at potentials comprised between the theoretical oxygen evolution and the electrochemical activity of NiMoO₄. What is the reason for suppressing oxygen evolution? This point does not seem to be addressed in the literature, but would be helpful to better understand asymmetric configurations with NiMOO₄ and assess their viability.

Our Response: We are thankful to the reviewer for the very helpful suggestions. Oxygen evolution does theoretically happen at the potential around 0.163 V vs. SCE in 2 M KOH electrolyte.

In practice, the oxygen evolution reaction cannot be initiated at the theoretical point due to “polarization”, and overpotential always exists. We have tested the polarization curve of NiMoO₄ in 2M KOH water solution at a relatively low scan rate of 1 mVs⁻¹ (new Supplementary Fig. 8d-f)

(d) Charge curve of the NiMoO₄ nanowires at the current density of 1 mA cm⁻² in 2 M KOH electrolyte. (e) CV curve of the NiMoO₄ nanowires at a scan rate of 1 mV s⁻¹ in 2 M KOH electrolyte. (f) Water oxidation potential of the NiMoO₄ nanowires at the scan rate of 1 mV s⁻¹ in 2 M KOH electrolyte.

From the linear sweep voltammetry (LSV) analysis, it can be found that oxygen evolution starts at ~0.520 V vs. SCE in the NiMoO₄ electrode i.e. the overpotential is 0.357 V (compared with 0.163 V). Thus, it is safe for NiMoO₄ to cycle (at the scan rate of 1 to 10 mV s⁻¹) between 0 and 0.5 V vs. SCE without oxygen evolution.

Oxygen evolution reaction is indeed a kinetically sluggish process with high overpotential. Reaction processes in basic solution are shown as follows (*Angew. Chem. Int. Edit.* 2016, 55, 3694):

The reaction includes multistep proton-coupled electron transfer and the formation of O-O bond (*J. Electroanal. Chem.* 2011, 660, 254; *Chem. Soc. Rev.* 2015, 44, 2060). Thus, large polarization is inevitable for oxygen evolution and catalysts with high efficiency are required. NiMoO₄ is not a highly efficient catalyst for oxygen evolution. Based on the above reasons, the NiMoO₄ electrode can work in the potential range of 0–0.5 V vs. SCE without oxygen evolution.

Finally, the stability of NiMoO₄ nanowires in the potential range of 0–0.5 V vs. SCE in 2 M KOH electrolyte is also consistent with previous literatures (*Nano Energy* 2014, 8, 174–182; *ACS Appl. Mater. Interfaces* 2013, 5, 12905–12910; *J. Mater. Chem. A* 2015, 3, 22081-22087).

The following descriptions have been added in the revised manuscript and Supplementary Information.

"From the linear sweep voltammetry (LSV) analysis (Supplementary Fig. 8f), it can be observed that oxygen evolution starts at ~0.52 V vs. SCE in the NiMoO₄ electrode. Thus, it is safe for NiMoO₄ to be cycled between 0 and 0.5 V vs. SCE. "

"The NiMoO₄ electrode works in a potential range (up to 0.5 V) exceeding the theoretical oxygen evolution potential in 2 M KOH (~0.163 V). This may be attributed to the kinetically-sluggish oxygen evolution reaction (OER) process. In addition, NiMoO₄ is not an efficient OER catalyst. Hence, oxygen evolution does not occur at the theoretical potential due to polarization. "

ii) The trend in Fig. 3d (“Volumetric capacitance of the FeOOH nanoparticle anode at different mass loadings”) may look misleading since the volumetric capacitance of materials is known to decrease with the higher mass loading. This is obviously because the Authors calculate the volumetric capacitance of electrodes including the volume of current collectors. Instead, the volumetric capacitance of active material alone can be presented for the better clarity of data presentation since the volumetric capacitance of the total device (including all of its components) is also provided in the manuscript. Then, the common trend of lower volumetric capacitance for thicker electrode materials can be evidenced and will not delude the reader. Alternatively, the Captions of Fig. 3d can be modified to say explicitly that the volumetric capacitance includes the volume of current collectors, but providing the common trend is still the better option.

Our Response: We thank the reviewer for the very valuable comments. Based on reviewer #1’s very helpful suggestions, we have modified the captions of Figure 3d to state explicitly that the volumetric capacitance of the FeOOH electrode at different mass loadings includes the volume

of the current collector not to mislead the readership of our manuscript. Other changes have also been carried out in the revised manuscript to address this issue.

Based on Reviewer#1's helpful suggestions, the caption of Figure 3d has been modified as shown below.

Figure 3 | Electrochemical performance of FeOOH nanoparticle anode. (a) CV curves (b) Specific gravimetric capacitance as a function of current density (c) Specific gravimetric and area capacitances of the FeOOH nanoparticle anode at different mass loadings. (d) Volumetric capacitance of the FeOOH nanoparticle anode (including the volume of the current collectors) at different mass loadings. (e) Cycling performance of the FeOOH nanoparticle anode at 1.6 and 9.1 mg cm⁻². (f) Nyquist plot after 1st and 10000th cycle.

iii) As pointed out with respect to the initial submission, the positive electrode is not pseudocapacitive. Therefore, any mentioning of its capacitance is not relevant. To be technically rigorous, the Authors need to make corresponding changes throughout the manuscript. For example, formula (4) in the revised manuscript (and the text and formula below the Supplementary Fig. 9) requires corrections in the denominator by substituting $C\Delta V$ by the capacity of the NiMOO₄ cathode. Please also check the remainder of the text to make sure the capacitance of the positive electrode is not mentioned.

Our Response: We thank Reviewer #1 for his/her valuable suggestions. Based on the reviewer's valuable suggestions, we have checked thoroughly the revised manuscript and made the corresponding changes. The revised formula 4 in the manuscript is shown below.

"The mass ratio of the positive to negative electrode is obtained by using the equation below.

$$\frac{m_+}{m_-} = \frac{(C_- \cdot \Delta V_-)}{(C)} \quad (4)$$

where m_+ and m_- are the mass loading of the NiMoO₄ and FeOOH electrodes, respectively, C_- is the specific capacitance of the FeOOH electrode. ΔV_- is the potential window of the FeOOH electrode and C is the specific capacity of the NiMoO₄ electrode. "

In addition, the text underneath **new Supplementary Fig. 10** has been completely revised and rewritten as depicted below.

"For a hybrid supercapacitor, the mass balance is determined as follows.

$$Q_c = Q_b \quad (1)$$

$$Q_c = m_c C_s V_c \quad (2)$$

$$Q_b = m_b C_b \quad (3)$$

Substituting equations 2 and 3 into equations 1

$$m_c C_s V_c = m_b C_b \quad (4)$$

$$\frac{m_c}{m_b} = \frac{C_b}{C_s V_c} \quad (5)$$

Where Q_c is the charge of the capacitor or pseudocapacitive electrode, Q_b is the charge of the battery-type electrode, m_c is the mass of the capacitor or pseudocapacitive electrode, C_s is the specific capacitance of the capacitor or pseudocapacitive electrode, V_c is the potential window of the capacitor or pseudocapacitive electrode, m_b is the mass of the battery-type electrode and C_b is the specific capacity of the battery-type electrode."

iv) To present clear and unambiguous background, the Authors are advised to clarify the introduction by stating explicitly that they restrict their discussion to AQUEOUS hybrid cells employing a battery-type cathode and a PSEUDOCapacitive anode. Otherwise, a few statements can be surprising to the reader familiar with the other types of hybrid configurations. For example, Page 2: i) “The design of ASC and HSCs result in high energy density and improved cyclability due to the contributions from the different charge storage mechanisms in an extended potential window (up to 2 V)^{28,29}”. The maximum voltage window of a lithium-ion capacitor working in organic electrolyte is about 3.8 V. Cycle life is also typically lower with asymmetric or hybrid systems than with fully symmetric double layer capacitors; ii) “To obtain full supercapacitors with high energy density, new pseudocapacitive anodes need to be explored to replace the low-capacitance carbon materials³³”. Hybrid lithium ion capacitors usually employ battery-type anodes having a higher capacity (graphite, non-porous hard carbons, LTO) than that of capacitive anodes.

Our Response: We are grateful to Reviewer#1. The introduction has been duly revised based on the reviewer’s suggestions. We have restricted our discussion to aqueous hybrid devices utilizing a battery-type cathode and a pseudocapacitive anode in the revised manuscript.

The following descriptions have been added to the introduction and their related references have also been included.

"Asymmetric and hybrid supercapacitors (HSC) have been extensively studied as a promising strategy to increase the energy density²⁰⁻²⁶. A typical HSC consists of both faradaic and capacitive electrodes^{12,27}. Their design result in high energy density due to the contributions from the different charge storage mechanisms and the operating potential window can be extended up to 2 V in aqueous electrolytes^{28,29}. In addition, faradaic cathode materials have been extensively studied resulting in the development of high-performance cathodes for aqueous supercapacitors^{20,21,30-32}. For instance, nickel based oxides have been explored due to their improved electronic conductivity and rich redox reactions, arising from the high electrochemical activity of Ni^{26,28,33,34}. Despite the high performance of these cathode materials, the maximum energy density of their hybrid cells in aqueous electrolytes is largely hindered by the low specific capacitance of commonly-used carbon anodes³⁵⁻³⁷. "

"To further evaluate the performance of the FeOOH nanoparticle anode for aqueous hybrid supercapacitors, we also designed the suitable battery-type cathode, nickel molybdate (NiMoO₄) using a hydrothermal method."

"36. Zheng, J. P. The limitations of energy density of battery/double-layer capacitor asymmetric cells. *J. Electrochem. Soc.* **150**, A484–A492 (2003).

37. Pell, W. G. & Conway, B. E. Peculiarities and requirements of asymmetric capacitor devices based on combination of capacitor and battery-type electrodes. *J. Power Sources* **136**, 334–345 (2004). "

Response to Reviewer #2:

Reviewer #2 (Remarks to author):

The authors have properly addressed the issues. I would now recommend the publication of this work on Nature Communications.

Our Response: We thank you very much for Reviewer 2's very valuable review and kind recommendation for the publication of our manuscript (NCOMMS-16-09875A).